# The cysteine-rich virulence factor NipA of *Arthrobotrys flagrans* interferes with cuticle integrity of *Caenorhabditis elegans*

Jennifer Emser[1], Nicole Wernet[1], Birgit Hetzer [2], Elke Wohlmann[1] & Reinhard Fischer [1]✉

Animals protect themself from microbial attacks by robust skins or a cuticle as in *Caenorhabditis elegans*. Nematode-trapping fungi, like *Arthrobotrys flagrans*, overcome the cuticle barrier and colonize the nematode body. While lytic enzymes are important for infection, small-secreted proteins (SSPs) without enzymatic activity, emerge as crucial virulence factors. Here, we characterized NipA (nematode induced protein) which *A. flagrans* secretes at the penetration site. In the absence of NipA, *A. flagrans* required more time to penetrate *C. elegans*. Heterologous expression of the fungal protein in the epidermis of *C. elegans* led to blister formation. NipA contains 13 cysteines, 12 of which are likely to form disulfide bridges, and the remaining cysteine was crucial for blister formation. We hypothesize that NipA interferes with cuticle integrity to facilitate fungal entry. Genome-wide expression analyses of *C. elegans* expressing NipA revealed mis-regulation of genes associated with extracellular matrix (ECM) maintenance and innate immunity.

The nematode cuticle is a highly specialized structure and plays an important role in the development and survival of *Caenorhabditis elegans*. This exoskeleton is shaping the nematode, mediates movement through connection to the muscles, and protects the animal from mechanical stress or microbial infections. The cuticle, primarily composed of collagens, is secreted by the underlying epidermal cells and undergoes renewal through molting at each larval stage, with varying collagen compositions at different stages[1,2]. Mutations in certain collagens or enzymes involved in collagen biosynthesis can have lethal consequences, like the deletion of the *bli-4* gene that encodes the subtilisin-like serin protease Kex2 and is involved in N-terminal processing of collagens[3]. Likewise, mutations in collagen-encoding genes like *dpy-7* or *bli-2* can give rise to distinctive body phenotypes, including the dumpy (Dpy) or the blister (Bli) phenotype, respectively[4]. Mutations in orthologous genes in humans can lead to various diseases, including *osteogenesis imperfecta* or *epidermolysis bullosa*[5].

The nematode-trapping fungus *Arthrobotrys flagrans* (formerly *Duddingtonia flagrans*) senses the presence of nematodes through nematode-specific ascarosides and forms sophisticated adhesive ring-like networks[6–8]. In the absence of nematodes, trap formation is inhibited by fungal arthrosporols, and small volatiles lure nematodes into the mycelia and into the traps[9]. If high numbers of nematodes are present, ascarosides lead to downregulation of arthrosporols, thereby de-repressing trap formation. Recently, the fungal receptor for ascarosides has been characterized as a G-protein coupled receptor protein[7,8]. The complicated interplay between small signaling molecules prevents induction of traps if only a single nematode approaches a fungal colony[10–12]. The extracellular low-molecular weight compounds lead to reprogramming of the trap cells and the rearrangement of the cytoskeleton[7,8,13–15]. Ultimately, cell-to-cell communication is a prerequisite for ring closure[16,17].

The first step in the interaction of *A. flagrans* and a nematode after capture is the formation of a penetration hypha. This specialized structure exerts pressure and secretes lytic enzymes, thereby enabling the fungus to overcome the nematode's cuticle and epidermis[18]. Afterwards an infection bulb is formed inside the nematode body from

[1]Institute for Applied Biosciences. Department of Microbiology, Karlsruhe Institute of Technology (KIT) - South Campus, Fritz-Haber-Weg 4, Karlsruhe 76131, Germany. [2]Max Rubner-Institut (MRI) - Federal Research Institute of Nutrition and Food, Haid-und-Neu-Strasse 9, Karlsruhe 76131, Germany. ✉e-mail: reinhard.fischer@KIT.edu

where trophic hyphae grow out to colonize and dissolve the entire animal[19,20]. It was assumed that penetration and digestion only rely on secreted fungal enzymes, which dissolve the cuticle, proteins, nucleic acids, and other polymers. However, the genome of *A. flagrans* encodes more than 200 small-secreted proteins (SSPs), more than 100 of which were predicted as putative effector proteins[19]. Analysis of RNAseq data of other nematode-trapping fungi (NTF) and the genome of *A. flagrans* revealed that many of the upregulated SSPs might be involved in the attack since they are found in the PHI (pathogen-host interaction) database. Additionally, many of them are bioinformatically predicted to be involved in infections (EffectorP3.0). Indeed, a recent study in *A. flagrans* described the importance for one of such SSPs, the virulence factor CyrA (cysteine-rich protein A). CyrA is important for paralysis of the nematode during the infection process[20]. This study paved the way for the analysis of other SSPs in fungal-nematode interactions.

In other host-pathogen interactions, the utilization of virulence factors, effector proteins, or SSPs are used to suppress host immune reactions and allow colonization[21,22]. The oomycete *Phytophthora infestans* uses secreted effector proteins to repress plant defense mechanisms[23]. In the maize smut fungus *Ustilago maydis*, the virulence factor Rsp3 is displayed on the surface of the fungus. This helps *U. maydis* to effectively evade recognition and subsequent activation of plant defense responses and enables successful penetration and establishment of infection within the host[24]. Due to long co-evolutionary history between hosts and pathogens, host-pathogen interactions often involve an ongoing arms race resulting in the evolution of new effectors and new defense mechanisms[25]. Hence, the deletion of individual effector proteins typically has minimal or no significant impact on the overall virulence of the pathogen. Nevertheless, many effector or virulence proteins target host processes for which they may have to enter the host cells. Whereas bacteria use e.g., the type III-secretion system for injection of proteins into the cells, the mechanisms in fungal infections are largely unknown[26]. Recently, it was discovered that several effector proteins of *U. maydis* form a complex at the fungal membrane. The complex plays a critical role in host cell penetration and interacts with plant plasma membrane ATPases potentially facilitating the transport of other effector proteins[27]. In the rice blast fungus *Magnaporthe oryzae* some effectors are internalized by endocytosis[28].

Here we characterized a small-secreted effector protein, NipA (nematode-induced protein A) of *A. flagrans*. *nipA* expression was upregulated during infection of *C. elegans*, and the protein accumulated at the penetration site. Interestingly, NipA induced the formation of blisters when heterologously expressed in epidermal cells. This suggests a role in the early stages of nematode infection.

## Results

### NipA is a cysteine-rich, small-secreted protein

Analysis of the *A. flagrans* secretome revealed an extensive array of SSPs with a potential role in virulence against nematodes[19]. As a first indication for a putative function of such SSPs in fungal-nematode interactions, we checked whether they were predicted to act as virulence or effector proteins (EffectorP[29]), whether they were unique to nematophagous fungi, and whether they were upregulated during the predatory lifestyle (published RNAseq data). Meeting these criteria with one of the highest EffectorP hits in *A. flagrans* with an effector probability of 0.95, was NipA. The NipA protein consists of 133 amino acids and harbors a signal peptide at its N-terminus (Amino acids (AA) 1-16). Orthologous proteins were only found in the closely related fungus *Arthrobotrys oligospora*. The sequence of NipA contains thirteen cysteines, and analysis by ProtNLM predicted similarity to EGF-domain containing proteins. AlphaFold predicted six disulfide bridges and one reduced cysteine at the N-terminus (Fig. 1a)[30]. To test the functionality of the predicted signal peptide, the laccase C protein of *Aspergillus nidulans* lacking its own signal peptide was C-terminally

fused to NipA and expressed under the control of the constitutive *gpdA* promoter of *A. nidulans*. The transformed *A. flagrans* strain was grown on media with 1 mM ABTS. Secreted laccase oxidizes ABTS to a more stable cation and results in a color change of the medium. After 48 h incubation at 28 °C, the color of the medium around the hyphae turned blue. Colonies of wild type or of a strain with laccase fused to NipA without signal peptide did not show any coloration. These results suggest that the predicted NipA signal peptide is functional (Fig. 1b).

Co-localization studies using green fluorescent protein (GFP)-labeled Lifeact revealed striking similarity in the distribution patterns of NipA-mCherry and Lifeact-GFP within patches in empty traps. These results suggests that NipA is actively transported in vesicles along the actin cytoskeleton (Fig. 1c)[15].

### *nipA* is highly upregulated during the infection and accumulates at the penetration site

Another criterium for the action of SSPs, besides their secretion, is their stage-specific expression. Differential expression of *nipA* was investigated by quantitative real-time PCR (qRT PCR) using RNA from uninduced vegetative mycelium (Ctrl.) and mycelium-containing traps after co-incubation with *C. elegans* (Ind.). The relative expression of *nipA* in uninduced mycelium was at $4.3 \times 10^{-5}$ ($\pm 5.1 \times 10^{-6}$ SD; $n = 3$) while the relative expression in mycelium induced with nematodes was at 4.5 ($\pm 0.2$ SD; $n = 3$) ($p$-value < 0.0001; ****). The expression was normalized to *A. flagrans* actin (dfl_002353) (Fig. 2a).

The analysis of the *nipA*-transcript level by qRT PCR did not allow to distinguish between transcriptional upregulation in vegetative hyphae and in fungal traps because all fungal material from induced plates was pooled for RNA extraction. Therefore, a promoter fusion assay was used to resolve the spatial and temporal expression of *nipA* microscopically in hyphae[15]. The *histone 2b* gene was fused to mCherry and expressed under the control of the *nipA* promoter. The fusion protein accumulates in nuclei due to nuclear localization of the histone protein. The constitutive *h2b* promoter was used as a control, expressing *h2b-GFP*. *A. flagrans* was co-transformed with both constructs. Traps of the respective strains were induced by co-incubation with *C. elegans* on LNA-microscopy slides for 24 h at 28 °C. Strong mCherry signals were observed in nuclei of the trap hyphae (Fig. 2d arrowheads) but barely in vegetative or trophic hyphae inside *C. elegans*. Fluorescence decreased in mycelium further away from the traps and was only hardly visible in areas without traps and in the uninduced mycelium (Fig. 2d). The fluorescence intensity of the vegetative mycelium and traps expressing GFP under control of the *h2b* promoter was comparable in all nuclei (control = 85.44, ±16.6 SD; induced = 81.03, ±24.36 SD). The total cell fluorescence of GFP and mCherry of single nuclei of induced (Fig. 2c) and uninduced (control) (Fig. 2b) samples was quantified and compared. The average fluorescence intensity of mCherry (8.849, ±2.48 SD) in uninduced mycelia was significantly lower than the GFP signal (85.44, ±16.6 SD) ($n = 350$; $p$-value < 0.0001; ****). However, in traps the mCherry fluorescence intensity was significantly enhanced (190.5, ±1134.1 SD), compared to the GFP intensity (81.03, ±24.36 SD) ($n = 350$; $p$-value < 0.0001; ****) (Fig. 2b, c).

Next, we fused the NipA protein to GFP or mCherry and expressed the fusion proteins under the control of the *nipA* promoter in *A. flagrans*. No signal was detected in uninduced mycelium, while empty traps of induced hyphae showed vesicle-like structures that accumulated at the inner rim of the traps (Fig. 3a). Upon infection of nematodes, NipA localized in a disk-like shape at the contact site between penetration peg and the nematode which is the site of later intrusion. During the formation of the penetration hyphae, NipA formed a ring around the entry point (Fig. 3c). Co-localization of NipA-mCherry with Lifeact-GFP showed an overlay of the signals in traps (Fig. 1c), as did co-localization with CyrA, a previously described virulence factor, in empty traps (Fig. 3b). However, as the infection progressed, a change in the localization of both effector proteins was observed. While CyrA

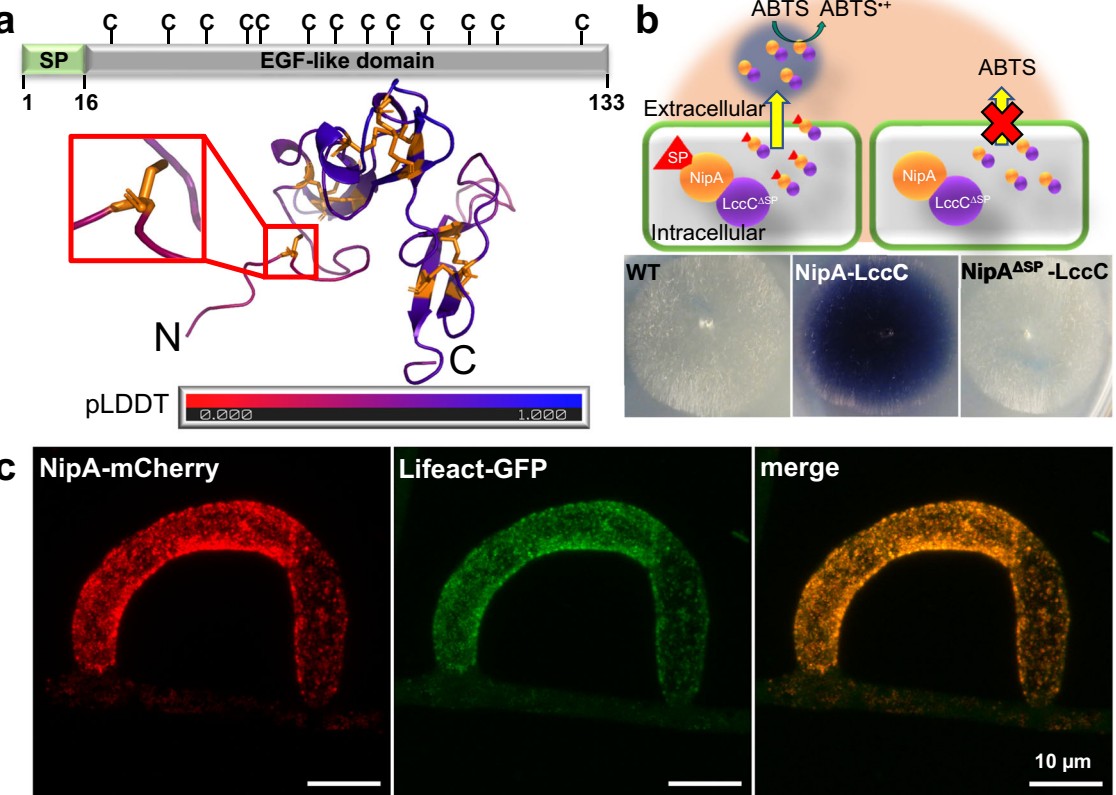

**Fig. 1 | NipA is a small-secreted protein with an EGF-like domain. a** NipA contains a 16 amino acid long predicted SP and 13 cysteine residues scattered along the protein. According to AlphaFold[30] (pLDDT of 61.32) structural predictions the first cysteine at position 23 does not form an intramolecular disulfide bridge. **b** Scheme of the Laccase-Assay. NipA was C-terminally fused to LccC from *A. nidulans* lacking its own signal peptide and expressed under control of the constitutive *gpdA* promoter. Secretion of the laccase is driven by the SP of NipA and leads to the oxidation of ABTS to a more stable ABTS-cation resulting in a blue color change in the medium. Deletion of the NipA-SP prevented the NipA-Laccase C fusion protein from being secreted (upper panel). The wild-type strain and transformants expressing the full-length *nipA-lccC^ΔSP* or *nipA^ΔSP-lccC^ΔSP* constructs were plated on PDA plates containing 1 mM ABTS and incubated for 48 h at 28 °C (lower panel). **c** Co-localization of NipA-mCherry and LifeAct-GFP. sVWZ was transformed with pJM16 and selected on geneticin-containing PDA-plates. For microscopic analyses spores of positive transformants were transferred onto thin LNA-slides and co-incubated with a mixed population of *C. elegans* for trap induction. Images were obtained using a confocal microscope with an AiryScan module.

accumulated in the infection bulb beneath the nematode's cuticle, NipA remained at the penetration site outside the nematode (Fig. 3d arrow). The fusion protein was not visible inside *C. elegans*. This suggests that NipA is involved in early stages of the infection while CyrA plays a role in later stages.

**Deletion of *nipA* leads to slower penetration of the nematode**

To determine the molecular function of NipA in the infection process, we created an *A. flagrans nipA*-deletion strain by replacing the *nipA*-open reading frame (ORF) with the hygromycin resistance cassette via homologous recombination. Transformants were selected on PDA-plates containing hygromycin, and replacement of the *nipA* ORF was verified by PCR and Southern blot analysis (Fig. 4a, b). Phenotypically the *nipA*-deletion strain was indistinguishable from wild type when grown on PDA. The strain grew with the same growth rate as wild type (Fig. 4c). Because NipA accumulated at the fungal entry point into *C. elegans*, we anticipated a function during penetration and hypothesized that the penetration time could be prolonged if NipA was absent. To test this hypothesis, we developed CyrA-mCherry as a tool to visualize the penetration according to its accumulation at the infection bulb beneath the epidermis. Therefore, CyrA-mCherry was expressed under the native *cyrA* promoter in wild type and the *nipA*-deletion background. Spores of the transformed strains were plated on thin LNA-agar pads and mixed populations of *C. elegans* were added to each strain and incubated for 24 h at 28 °C to induce trap formation.

Subsequently, uncaptured nematodes were washed off with H₂O, and areas with traps were transferred into an eight-well IBDI chamber. Just before recording, synchronized young adult *C. elegans* were added to the samples. Time was measured from capturing a nematode until the beginning of CyrA-mCherry accumulation and thus successful penetration and bulb formation (Fig. 4e). Recordings were made every nine min for a period of 11 h. The penetration time of the *nipA*-deletion strain was significantly delayed (77.53 min ± 14.07 SD; *n* = 45; *p*-value < 0.0001; ****) compared to the wild-type (64.18 min ± 9.92 min; *n* = 50). The mutant phenotype was rescued with a NipA-GFP version, proving that the GFP tag did not interfere with the biological function of the protein (63.81 min ± 10.39 min; *n* = 26; *p*-value = 0.8790; ns). In contrast, NipA-GFP in which cysteine 23 was changed to alanine was not functional (83.48 min ± 13.61 min SD; *n* = 21; *p*-value < 0.0001; ****). To reduce outliers, we used the 20% trimmed average of the measured times (Fig. 4d). Significant differences to the wild type were calculated using the unpaired *student's t-test*. The delayed penetration of the *nipA* deletion mutant indicates a role of NipA during early stages of the interaction.

**Heterologous expression of NipA results in developmental defects in *C. elegans* larvae**

Another tool for the analysis of the molecular function of effector molecules is the heterologous expression of the candidate protein in the host organism. This method potentially allows the magnification of the

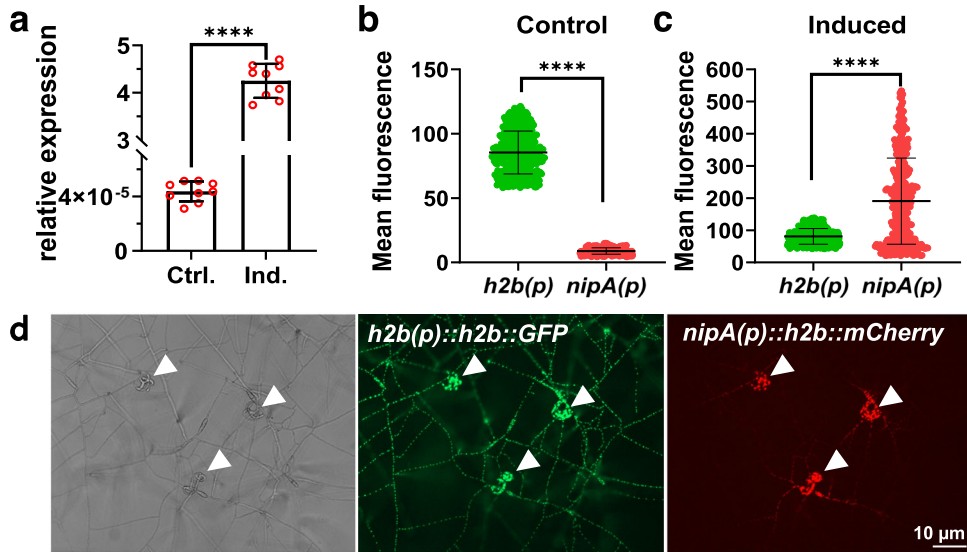

**Fig. 2 | *nipA* expression is upregulated during infection. a** qRT-PCR analysis of *nipA* expression in uninduced (Ctrl.) and induced (Ind.) mycelium. For uninduced samples, $10^6$ spores were plated onto LNM plates covered with cellophane and were incubated for 48 h at 28 °C. For induced mycelium a mixed *C. elegans* population was added after 24 h and co-incubated for another 24 h at 28 °C. RNA was isolated using Trizol. Three biological replicates with three technical replicate each, are displayed. Mean value (Ctrl.) = $4.377 \times 10^{-5} \pm 6.242 \times 10^{-6}$ SD and mean value (Ind.) = $4.25 \pm 0.322$ SD. Expression was normalized to actin. **b–d** Quantification of fluorescence intensities of the promoter reporter assay (*nipA(p)::h2b::mCherry*; red values in columns) in uninduced (**b**) and in induced mycelium (**c**) from microscopic images (**d**) using ImageJ. All nuclei were labeled by GFP (*h2b* driven expression of *h2b-GFP*) (green values in columns). Grey-values of nuclei were calculated in the GFP and RFP channels relative to the background. Arrow heads indicate trapping sites. Data are presented as mean values ± SD. **b** Mean value (*h2b(p)*) = 85.42 ± 16.59 SD with a minimum of 58.13 and a maximum of 121.2. Mean value (*nipA(p)*) = 8.849 ± 2.477 SD with a minimum of 4.438 and a maximum of 14.88. **c** Mean value (*h2b(p)*) = 81.03 ± 24.36 SD with a minimum of 42.34 and a maximum of 138.6. Mean value (*nipA(p)*) = 190.5 ± 134.1 SD with a minimum of 21.18 and a maximum of 533.2. In order to compensate for measurement inaccuracies, the 12.5% trimmed mean was determined for (**b**) and (**c**). (**a–c**) Significance was calculated using unpaired two-sided *student's t-test* using a 95% confidence interval (*t*-test < 0.0001). ****$p < 0.0001$; ***$p < 0.005$; **$p < 0.01$; *$p < 0,05$; n.s., $p > 0.05$. Error bars indicate standard derivation. Source Data are provided as a Source Data file.

natural effect of the effector molecule by acting throughout the entire nematode body instead of a very local response at the infection site. To this end, *nipA* was expressed under different nematode-specific promoters in the *C. elegans* N2 wild-type strain. A NipA-mScarlet fusion protein was expressed as extrachromosomal array under the control of the *elongation factor 1-alpha* (*eft-3*) promoter, resulting in the expression of *nipA-mScarlet* in all tissues and developmental stages of the worm. As a control, just the fluorophore alone was expressed using the same promoter. Animals expressing NipA-mScarlet in the N2 wild-type background showed strong accumulation of fluorescence in the vacuoles of coelomocytes, while in the control strain the fluorescence was evenly distributed (Fig. 5a). The accumulation of NipA-mScarlet in the coelomocytes was verified by co-localization with free GFP expressed under the *unc-112* promoter as well in the N2 background (Fig. 5c, d). Coelomocytes are scavenger cells that endocytose fluid from the pseudocoelom and have been implicated in *C. elegans* immunity by a proposed detoxification function[31]. Because they endocytose proteins from the body cavity the accumulation of NipA-mScarlet in these cells indicates degradation of the fusion proteins. Thus, to reduce degradation of NipA-mScarlet, the constructs were also expressed in GS2478, which is defective in endocytosis by coelomocytes because of a mutation in the *cup-8* gene, and thus uptake of proteins is less efficient. NipA-mScarlet still localized in the coelomycetes of older larvae and adults, albeit somewhat weaker than in wild type (Fig. 5b) probably due to the fact that this mutant is not fully coelomocyte deficient. A viability assay of the *nipA*-expressing strain revealed an increased number of unhatched eggs and deformed larvae as compared to the mScarlet-expressing controls (Fig. 5e).

Although expression of NipA caused an observable phenotype, the results have to be taken with caution because in the natural infection process NipA is probably only in contact with *C. elegans* cells early during penetration and not in all tissues during larval development. Nevertheless, this observation indicates that the NipA protein affects the nematodes. Therefore, NipA-mScarlet was subsequently expressed as an extrachromosomal array under the control of the hypodermal *dpy-7* promoter in *C. elegans* N2. As a control *mScarlet* alone was expressed. In both strains, *mScarlet* signals were visible in the epidermis. Screening for body phenotypes revealed the appearance of abnormal shaped larvae in the *mScarlet*-expressing strain and deformed larvae in the *nipA-mScarlet* expressing strain. A viability assay revealed reduced viability of transgenic eggs with strong fluorescence in both strains. 17.78 % (±3.85% SD; *n* = 95) of the hatched *nipA-mScarlet* expressing larvae displayed a similar deformed phenotype as observed under the *eft-3* promoter, while only 5.35% (±2.35% SD; *n* = 110) of the control larvae were deformed (Fig. 5f). In both stains fluorescent eggs were observed which were not able to hatch. This might be due to cytotoxic effects of mScarlet overexpression[32]. Nevertheless, the observed reduced hatching rate and developmental phenotype in hatched larvae was higher than in the control, indicating a toxic effect of NipA on nematodes during development (*p*-value = 0.008; **). CyrA expression did not cause those phenotypes.

Taken together, we show that expression of *nipA* is harmful to *C. elegans* and affects larval development when expressed at early developmental stages, regardless of the expressing tissue.

## NipA causes blister formation in *C. elegans*

Nematodes mold between developmental stages, replacing their old cuticle with a new one. The composition of the collagens and the structure of the cuticle differs in every phase of their life. To test if NipA influences the cuticle itself, full-length NipA was fused to mScarlet and expressed under the *col-19* promoter, which leads to expression in the epidermis of adult nematodes and is thus not lost through molting and should not have an impact on the early larval stages. Experiments were carried out in N2 wild type and GS2478

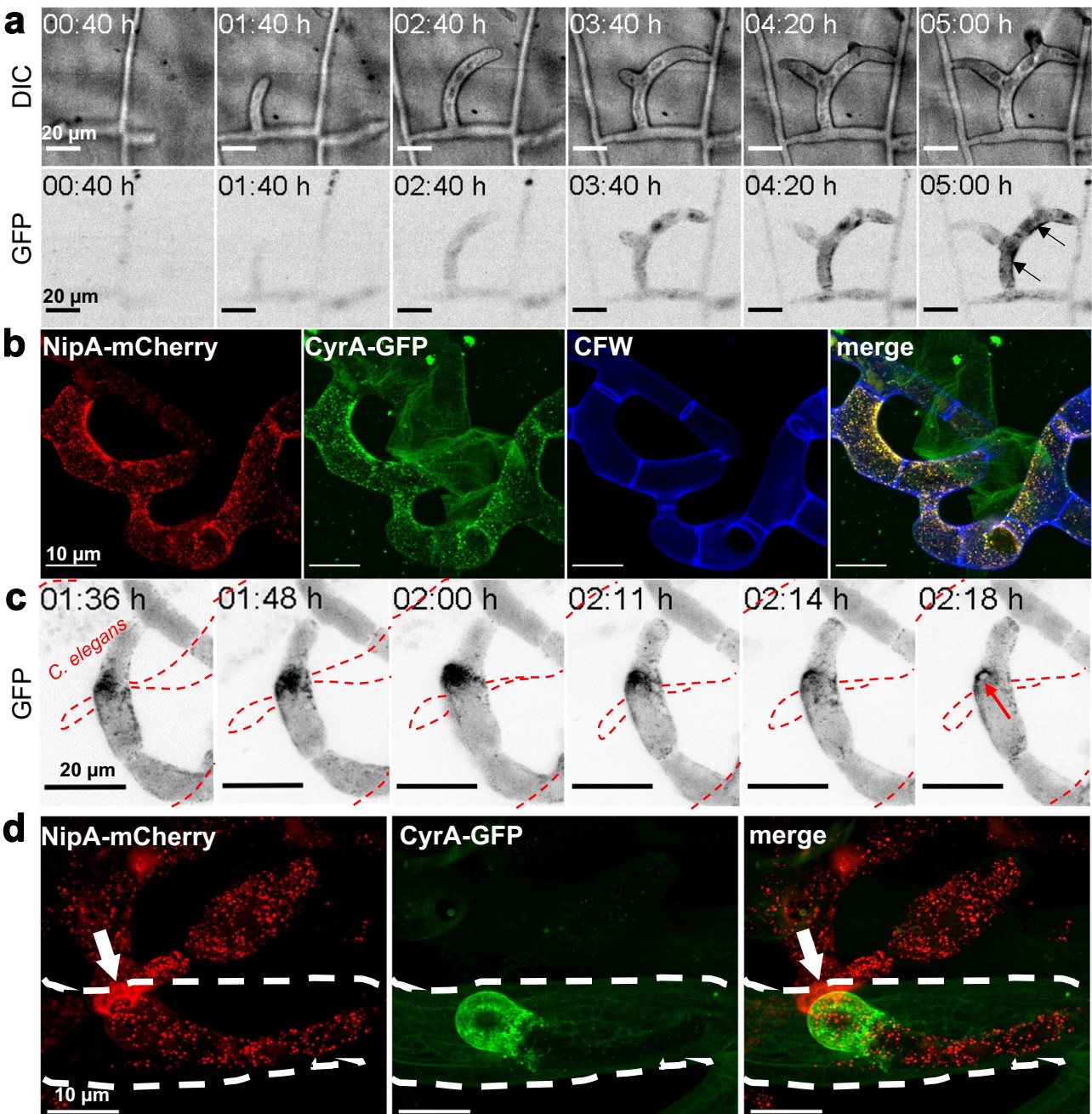

**Fig. 3 | NipA localizes at infection sites.** For trap induction spores were co-incubated with *C. elegans* for 24 h at 28 °C. **a** Observation of trap formation in the *nipA-GFP* expressing reporter strain over a time span of 5 h using a confocal microscope. Pictures were obtained every two minutes in both the DIC (upper panel) and the GFP (lower panel) channel. NipA-GFP started accumulating in the hyphae of induced mycelium before ring closure. Arrows indicate NipA-GFP localization at the inner rim of a closed trap. **b** Co-localization of NipA-mCherry and CyrA-GFP in empty traps. Both proteins localize at the inner rim of the traps. Cell walls are stained with Calcofluor white. **c** Time-course observation of NipA-GFP translocation during the infection process. NipA-GFP is actively transported to a specific contact site with a trapped nematode (red dotted line). It accumulates in one spot before localizing in a pore-like ring at the penetration site (red arrow). **d** Co-localization of NipA-mCherry and CyrA-GFP during the infection process. NipA-mCherry localized at the outer infection site while CyrA-GFP accumulated in the infection bulb. Arrow indicates direction of penetration. Dotted line marks outline of the captured nematode.

background. *mScarlet* alone was expressed using the same promoter. Three days after reaching adulthood some nematodes expressing *nipA-mScarlet* displayed brightly fluorescent blister-like structures spreading across their bodies (Fig. 6a). Opening of these blisters with a needle resulted in leakage of fluorescent liquid into the medium, while the nematode suffered no further damage (Fig. 6b). This phenotype was observed in both, N2 and GS2478 backgrounds while it was not observed in strains only expressing mScarlet. Expression of

the virulence factor CyrA as GFP-fusion protein in the epidermis did not cause any blisters[20].

In order to explore if the NipA target locates intra- or extracellular of *C. elegans* cells, the signal peptide of NipA was deleted. This modified construct was then injected into N2. In this strain, blister formation was still possible but took longer than in the case of expression with the signal peptide. Interestingly, the liquid inside the blister was still fluorescent. This suggests that NipA might leak into the ECM

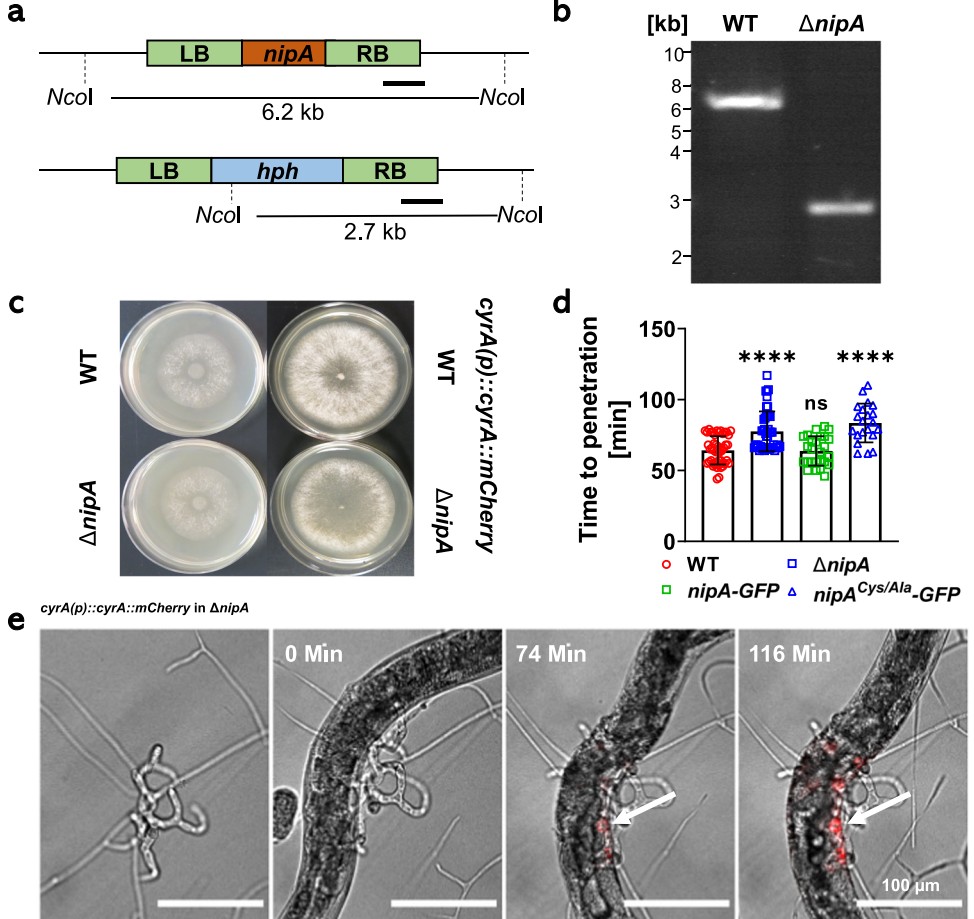

**Fig. 4 | Deletion of *nipA* delays penetration of *A. flagrans*. a** Scheme of the deletion strategy for the *nipA* gene. 1.2 kb of the *nipA* flanking sites were amplified from gDNA and cloned into the pJET1.2 vector flanking a hygromycin resistance cassette. *Nco*I restriction sites and resulting fragment sizes for Southern blot analysis are indicated. **b** Southern blot of DNA of wild type and the *nipA*-mutant strain. A 1 kb DIG-labeled probe binding to the 3′ flanking region was used (indicated as line in (**a**)). Expected bands at 6.2 kb for the wild type and at 2.7 kb for the *nipA*-mutant were obtained. **c** Growth comparison between *A. flagrans* wild type, *nipA* mutant, and wild type and the *nipA*-mutant strains expressing *cyrA-mCherry* under the native *cyrA*-promoter on PDA plates. **d** Penetration assay of CyrA-mCherry expressing wild-type (WT) (red circles; *n* = 50; mean value = 64.18 min ± 9.92 SD),

*nipA*-deletion strain (Δ*nipA*) (blue squares; *n* = 45; mean value = 77.53 min ± 14.07 SD), *nipA-GFP* re-complementation strain (green squares; *n* = 26; mean value = 63.81 min ± 10.39 SD) and re-complementation strain expressing mutated *nipA* (*nipA^Cys/Ala^-GFP*) (blue triangles; *n* = 21; mean value = 83.48 min ± 13.61 SD). Significances were calculated using the two-sided unpaired *student's t-test* using a 95% confidence interval ($p$ (WT vs. Δ*nipA*) < 0.0001, ****; $p$ (WT vs. *nipA-GFP*) = 0.879, ns; $p$ (WT vs. *nipA^Cys/Ala^-GFP*) < 0.0001, ****). Error bars indicate standard derivation. **e** Representative pictures from the penetration assay are displayed. Time measurement started at first trap contact and was stopped at the first accumulation of CyrA-mCherry signal inside the nematode body. Arrow indicates the infection site. Source Data are provided as a Source Data file.

(extracelluar matrix) even when expressed in epidermal cells without a signal peptide or the *mScarlet* is cleaved off (Fig. 6a). Thus, it remains unclear from these experiments if the target is intra- or extracellular.

To rule out any effect of the fluorophore, NipA was additionally expressed without *mScarlet* in wild-type background (Fig. 6c). Here, blistering was delayed by up to 10 days, suggesting a higher expression level of *nipA-mScarlet* or a positive effect of mScarlet on the phenotype. Animals expressing only *mScarlet* in wild-type and GS2478 background showed no blistering. To exclude that extensive *mScarlet* expression is inducing blister formation, we observed for 20 days 90 synchronized young adult nematodes of the control strains as well as nematodes expressing different NipA variants on 50 μM FUDR (5-Fluor-2′-Desoxyuridin) containing plates to suppress the formation of offspring. After 3 days the first blisters were formed in strains expressing NipA-mScarlet, while the N2 wild-type and the mScarlet expressing controls never showed blisters (Fig. 6d).

Collagen mutations can have an impact on the ultrastructure of the cuticle[33]. To verify if the ultrastructure is altered or blisters are

formed during the natural infection near the trap, Cryo-SEM images of captured nematodes were taken (Fig. 7a). Upon magnification of the penetration site, no apparent blisters were observed; instead, the nematode body appeared to be pushed inward where the penetration hypha made contact. Cryo-SEM imaging was also used to investigate if NipA influences the ultrastructure of the cuticle, or if blister formation alters it. Annuli and alae remained unaltered to the *C. elegans* wild-type. Nevertheless, it was observed that blister formation was restricted to areas without alae and stopped at the vulva (Fig. 7b). This can be explained by a different cuticle layer composition in these areas, missing the liquid-filled space in the medial layer[34].

Based on the AlphaFold structure prediction Cys-23 near the N-terminus remains unpaired and could therefore play an important role in the functionality of NipA, making it an interesting target for mutagenesis studies. To investigate this, we generated a *C. elegans* strain expressing an extrachromosomal array in which Cys-23 was replaced with alanine and expressed under the *col-19* promoter. Transgenic nematodes did not form any blisters, indicating an important function of Cys-23 in blister induction by NipA (Fig. 6d).

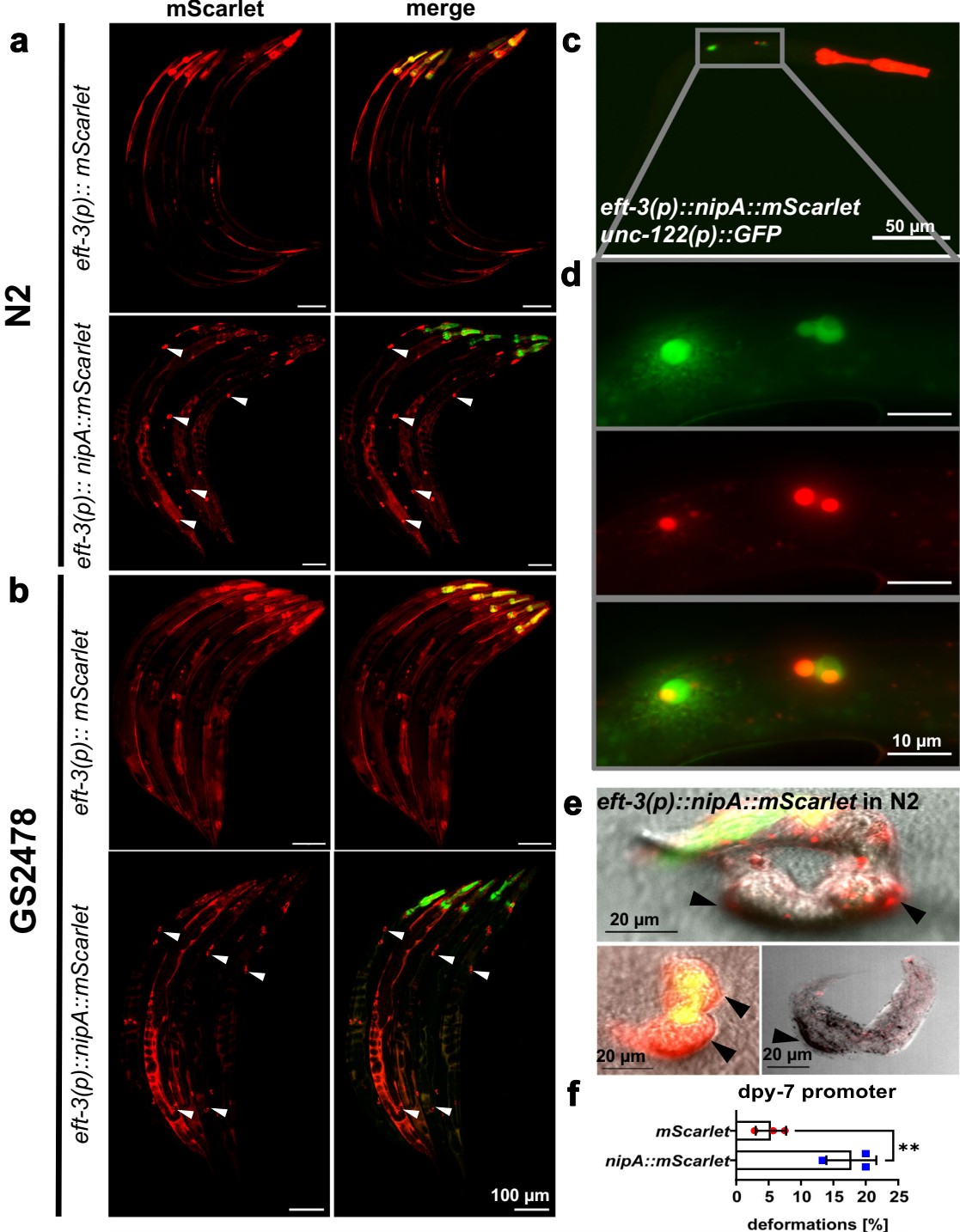

**Fig. 5 | Ubiquitous *nipA-mScarlet* expression under the *eft-3* promoter in *C. elegans* results in coelomocyte localization and deformed larvae. a** Localization of mScarlet and NipA-mScarlet in the N2 wild-type strain. Nematodes were synchronized and imaged at the young adult stage. Nematodes expressing *mScarlet* show signals mainly in the muscles and the epidermis. *nipA-mScarlet* expressing nematodes show accumulation at coelomycete resembling positions (arrowheads). **b** *mScarlet* and *nipA-mScarlet* expression in young adult nematodes of the coelomocyte defective strain GS2478. mScarlet localizes as in the N2 background. NipA-mScarlet localization remains in the coelomocyte resembling positions (arrowheads) whereby the signal appears weaker than in the wild-type background. **c** Micrograph of a N2 L1 larva expressing *nipA-mScarlet* under the *eft-3* promoter and *GFP* under the *unc-112* promoter for co-localization with coelomycetes. **d** Magnification of the area depicted in (**c**). Signals are partially overlapping (lower panel), indicating an accumulation of NipA-mScarlet in the vacuoles of the

coelomocytes. **e** Deformed larvae of *nipA-mScarlet* expressing wild type under the *eft-3* promoter. Transgenic young adult hermaphrodites were placed on fresh NGM plates with OP50 for 4 h to lay eggs. After 24 h hatched larvae were checked for viability. (**f**) Quantification of deformed larvae in *mScarlet* (red dots) and *nipA-mScarlet* (blue squares) expressing strains under the *dpy-7* promoter. Adult nematodes were transferred onto a fresh NGM-Plate with OP50 to lay eggs. Adults were removed after 4 h. Deformed larvae were scored after 2 days. Experiment was performed as technical triplicate with a total of 110 hatched nematodes from the control strain and a total of 95 nematodes from the *nipA* expressing strain, since each individual transformant is seen as biological replicate due to the extrachromosomal array expression. Data are presented as mean values ± SD. Significance was calculated using multiple *t-test* determining the statistical significance using the Holm-Sidak method (alpha = 0.05; *p*-value = 0.008826; **). Source Data are provided as a Source Data file.

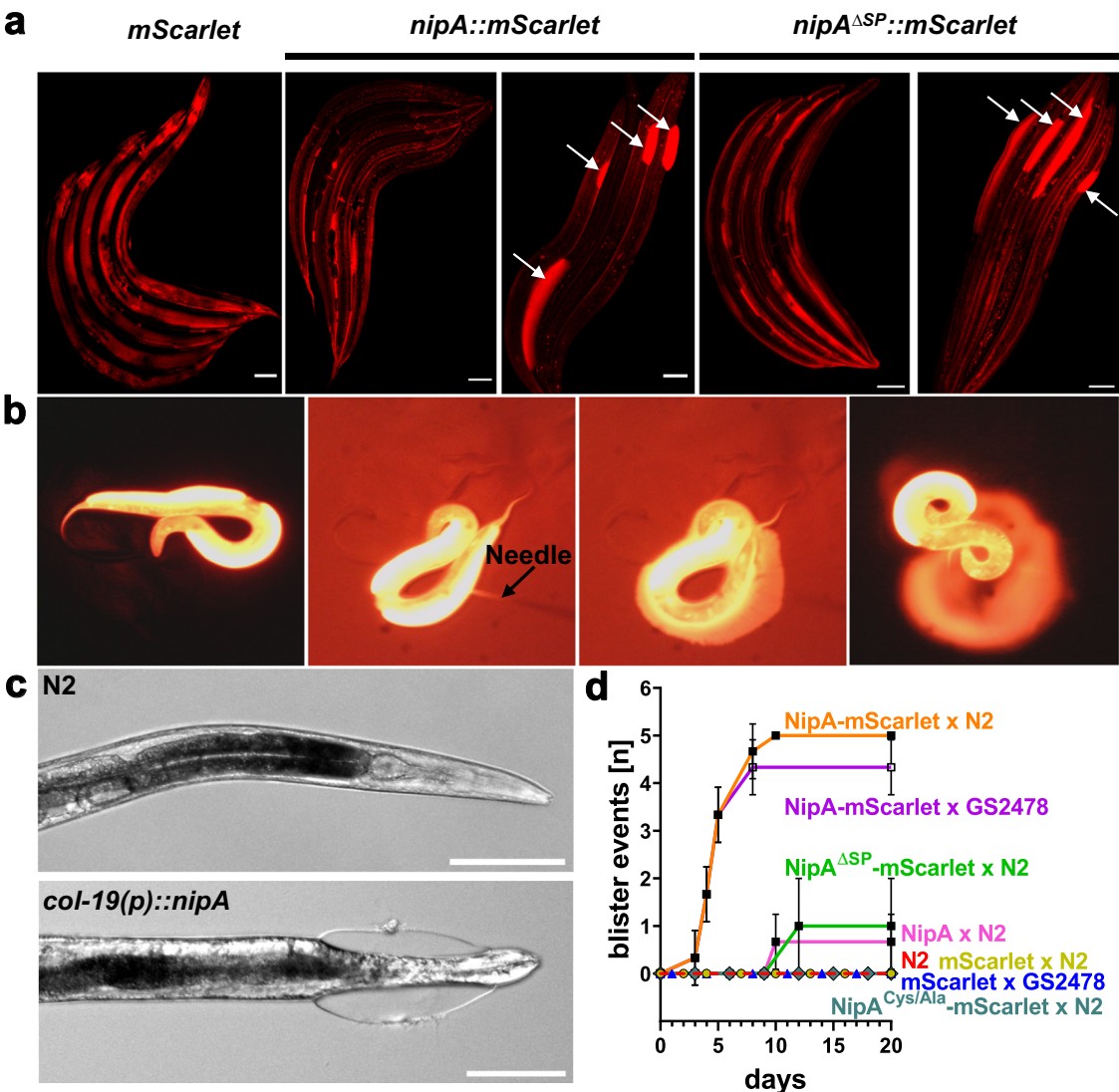

**Fig. 6 | Expression of *nipA* variants in the epidermis of adult nematodes under the *col-19* promoter leads to blister formation. a** Expression of *mScarlet* and *nipA-mScarlet* with and without signal peptide. Nematodes were synchronized and plated as L4 larvae onto NGM plates with OP50 containing 50 mM FuDR for long-term observations. Localization of the *mScarlet* signal in *nipA*-expressing strains was comparable to the control strain. Some individuals of both *nipA*-expressing strains formed brightly fluorescent blisters. **b** Opening fluorescent blisters with a needle resulted in leakage of fluorescent fluid onto the media without further harming of the nematode. **c** Expression of *nipA* without fluorophore is sufficient to induce blister formation. **d** Quantification of blister events in different strains expressing *nipA* variants. Animals were synchronized young adults on 50 μM FUDR-containing plates. Observations were carried out over 20 days. NipA-mScarlet induces blister formation in N2 wild-type and the GS2478 background after 3 days of adulthood (orange and purple lines, respectively). Without the signal peptide, blister formation occurred after 12 days (green). The *nipA* expressing wild type showed blistering after 10 days (pink). Nematodes expressing the cysteine mutated NipA variant showed no blister formation (dark green rhombus). Blistering was never observed in the *mScarlet*-expressing controls (yellow dots and blue triangles) or N2 wild type (red dotted line) (90 individuals per strain). Data are presented as mean values ± SD. Scale bar = 100 μm. Source Data are provided as a Source Data file.

## NipA induced blisters are formed within the cuticle

One reason for blister formation in adult nematodes is the disruption of epidermal cells, which is accompanied by an upregulation of *nlp-29*[35]. Therefore, we determined the *nlp-29* transcript level in *nipA*-expressing nematodes but found no difference in comparison to wild type. This suggested that NipA does not affect the basal membrane of the epidermis. To further strengthen this result, we investigated the presence of cellular components in the blisters and expressed *mScarlet* using the epidermal *dpy-7* promoter in *C. elegans* expressing untagged NipA under the control of the *col-19* promoter. The blisters of adult animals did not contain mScarlet (Fig. 8a).

Blister formation can also be caused by disturbance of the epidermis or cuticle integrity, or from mutations or deletions in proteins essential for cell-cell or cell-ECM contacts. Because the localization of proteins mediating cell-cell and cell-ECM contacts alter if the integrity of the membrane is disturbed, VAB-10A localization in blistered nematodes was investigated. VAB-10A is part of the fibrous organelles, localizing at the basal and apical membrane of the epidermis connecting the basal lamina and the cuticle via intermediate filaments[36]. Deletion or mutation of VAB-10A leads to blister formation. Disruption of the epidermal membranes or cells may lead to loss or mislocalization of VAB-10A[36,37]. Therefore, GOU2043, a VAB-10A-GFP expressing strain was transformed with *nipA-mScarlet* or only *nipA* under the control of the *col-19* promoter. The localization of VAB-10A-GFP did not change in *nipA*-expressing nematodes and was not altered in blistered areas (Fig. 8b). Taken together, our results suggest that the

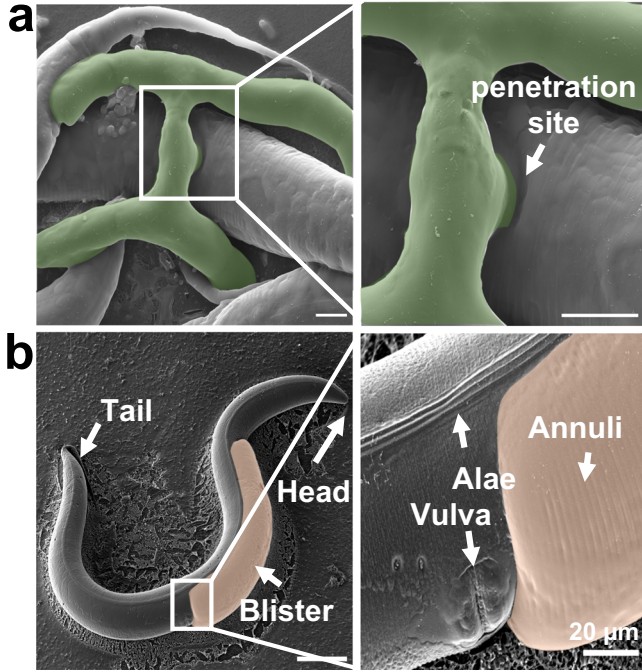

**Fig. 7 | Cryo-SEM analysis of the natural infection process and a *nipA* expressing nematode. a** *A. flagrans* hypha infecting young adult N2 nematode. Fungal mycelium was inoculated on thin LNA slides and co-incubated with *C. elegans* for 24 h. Synchronized young adult nematodes were added 1 h prior to microscopy. At the penetration site the nematode body is pushed inwards due to the pressure applied by the ingrowing hypha (scalebar = 5 μm). **b** SEM image of a nematode with a large blister caused by *nipA-mScarlet* expression. The blister is located at the dorsal anterior site (scalebar = 100 μm). The ultrastructure of the cuticle is visible and shows no significant difference to N2 (magnified panel on the right side). Blisters are restricted to areas without alae and stop at the vulva (scalebar = 20 μm).

blisters develop within the cuticle and the epidermis maintains its structural integrity.

## NipA alters the expression of ECM- and stress-response genes in *C. elegans*

Next, we performed genome-wide expression analyses to further understand the action of NipA in *C. elegans*. RNA of synchronized young adult nematodes of the *col-19(p)::mScarlet*, *col-19(p)::nipA*, *col-19(p)::nipA::mScarlet* and N2 wild-type strains was isolated using the TRIZOL method and sent for RNA sequencing. Comparison of the datasets revealed 238 differentially expressed genes in *nipA* and *nipA-mScarlet* expressing strains. Among these, 73 were upregulated, while 53 genes were downregulated (Fig. 9a). For gene ontology analyses of the up- or downregulated genes in both strains, the online tool WormCat was used[38]. Genes with a fold change >2 were considered upregulated and genes with a fold change <−2 were considered downregulated and used for the analysis. The downregulated genes predominantly fell into two categories: extracellular components such as collagen and stress response elements typified by C-type lectins. Conversely, the genes within the shared upregulated dataset predominantly aligned with stress response pathways associated with pathogen interactions and detoxification processes (Fig. 9b). The utilization of String-DB analysis allowed us to construct insightful interaction networks among the regulated genes. In the case of the upregulated gene set, CUB-domain-containing proteins, such as IRG-4, revealed a strongly connected network. Such proteins are involved in innate immunity and stress responses[39]. The network intricately links the immune-related entities with proteins involved in detoxification processes, such as UGT (UDP-glucuronosyltransferase). Among the downregulated genes, C-type lectin domain containing proteins appeared as a network. Another network interconnected collagen proteins, suggesting an orchestrated alteration of extracellular components (Fig. 9c). Due to the observed blister phenotype, we further analyzed the RNAseq data for differentially regulated genes potentially involved in blistering. Interestingly, among the downregulated collagen genes we observed significant downregulation of *bli-6* with a fold change of −2.4 (*p*-value = 3.5 × 10$^{-8}$) in the dataset of *mScarlet* versus

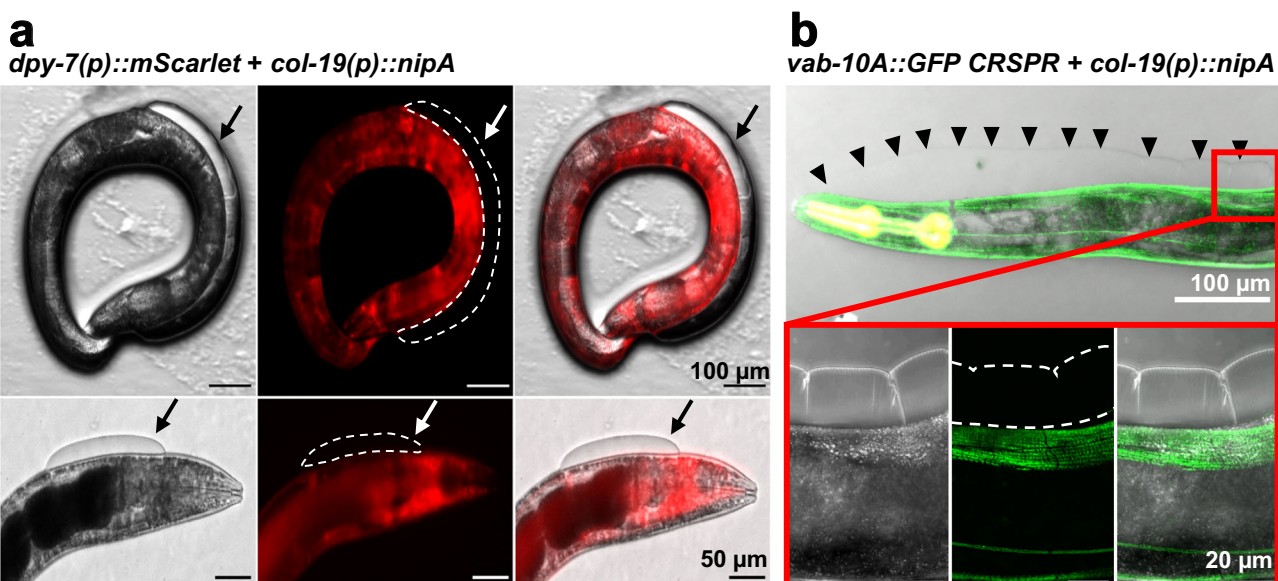

**Fig. 8 | NipA induced blisters are formed in the cuticle. a** Co-expression of *mScarlet* under the epidermal *dpy-7* promoter with *nipA* under the *col-19* promoter. NipA-induced blisters are not stained by epidermally expressed *mScarlet*, indicating no rupture of epidermal cells. Blisters are outlined in the RFP channel and indicated with an arrow. **b** Localization of VAB-10A-GFP in the epidermis of a *nipA*-expressing nematode. The localization pattern of VAB-10A is not affected by the formation of NipA induced blisters indicating no damage to the membranes of the epidermal cells. Arrow heads outline the edge of the blister.

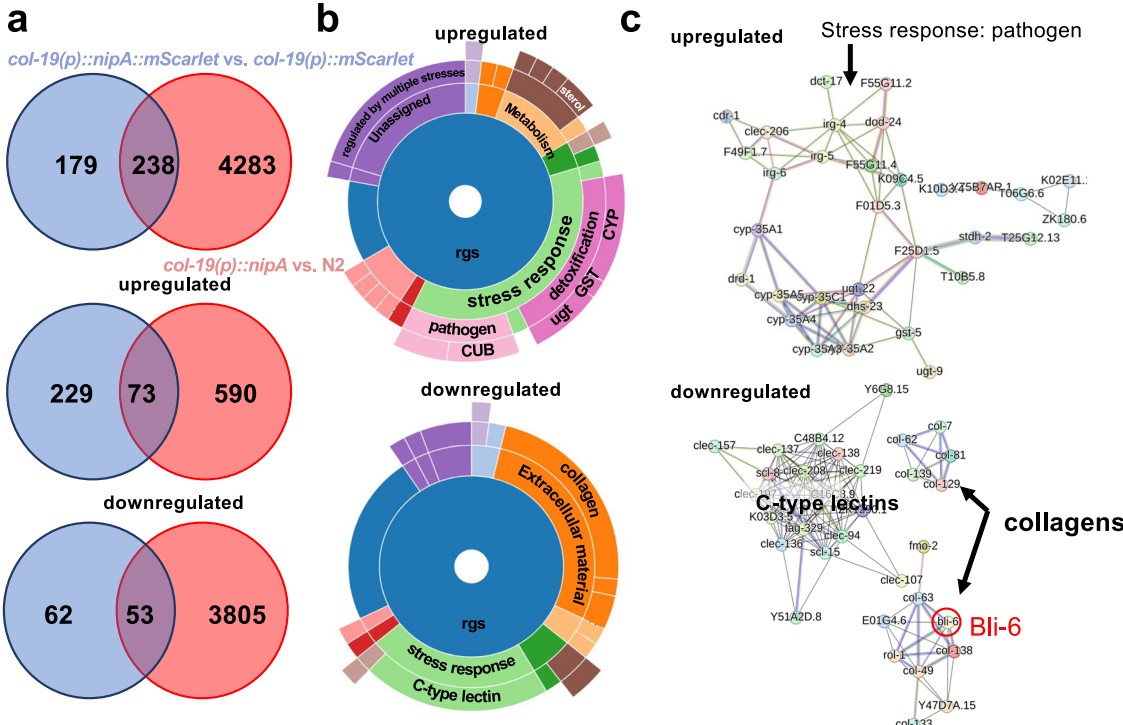

**Fig. 9 | Heterologous *nipA* expression induces transcription of detoxification genes and inhibits expression of collagen genes.** For RNAseq analyses RNA was extracted from strains expressing *nipA* and *nipA-mScarlet* under the *col-19* promoter and was compared to RNA from N2 wild type and *col-19(p)::mScarlet* expressing strains, respectively. **a** Venn-diagrams of differentially expressed genes in *nipA*-expressing strains. Red circle represents differentially regulated genes in the *nipA*-expressing strain compared to N2. Blue circle represents genes differentially regulated in the *nipA-mScarlet* expressing strain compared to the mScarlet control. Intersection area represents shared genes in both datasets. Upper graph shows the total number of regulated genes. *nipA* and *nipA-mScarlet* expressing strains share 238 differentially expressed genes. Among the upregulated genes a total of 73 genes are shared in both datasets, while 53 genes show the same downregulation in both strains. **b** Multi-level pie charts showing gene categories in common regulated genes (rgs) of the commonly up and downregulated gene sets created by WormCat. Genes with a fold change >2 were considered upregulated and genes with a fold change < −2 were considered downregulated and used for the analysis. Most of the upregulated genes are categorized as stress response genes for detoxification or pathogen defense. Many of the downregulated genes are classified as genes for ECM material or stress response, like collagens or C-type lectins respectively. **c** Protein interaction networks of the commonly regulated genes of both datasets created by String-DB. Proteins of the upregulated dataset show strong connections between each other, especially for proteins categorized for stress response. In the downregulated dataset strong interactions between the C-type lectin proteins are visible. Strong connections exist as well between different collagens with BLI-6 as one of the collagens with the highest number of connections.

*nipA-mScarlet* expressing strains and −3.4 (*p*-value = 1.96 × 10⁻¹⁰) in the N2 versus *nipA* expression dataset. In the String-DB network BLI-6 is connected to many other downregulated collagen genes. BLI-6 has been identified as a crucial component contributing to the structural integrity of the cuticle. Deletion or mutation of *bli-6* is consistently associated with blister formation[40,41]. Taken together, our results revealed that NipA might play a role in weakening the pathogen recognition and the outer barrier of the nematodes, thereby probably facilitating the infection process. In addition, it triggers parts of the innate immune response, whether as primary effect of NipA or as secondary effect by damage-triggered responses.

## Discussion

During fungal infection of nematodes, the cuticle serves as the first barrier encountered by the fungus. Immediately after penetrating the cuticle, the fungus reaches the monolayer of epidermal cells, which secrete the components of the cuticle[1]. The fungal protein NipA accumulated from outside at the penetration site while no translocation of the fusion protein into *C. elegans* cells was visible. Hence bioinformatic analysis with WoLF PSORT predicted NipA in the extracellular space of the host. In the absence of NipA, host penetration was slower than in wild type *A. flagrans*. These observations suggest a role for NipA during the penetration of *C. elegans* and interference with cuticle integrity.

To investigate the effect of NipA in the outer layers of *C. elegans*, different variants of *nipA* were heterologously expressed in the epidermis of adult nematodes. This expression along the entire nematode body resembles a magnifying glass, as it induces phenotypic changes all over the nematode body, which in the infection process are restricted to small areas or even individual cells. The most prominent effect of NipA expression was the formation of blisters in the cuticle of adult nematodes after a few days when expressed in the epidermal cells. Here the protein should be secreted because of the signal peptide. However, blistering was not prevented by deleting the signal peptide but delayed. During the natural infection process large blisters were not observed at the penetration site. It remains also open if blistering is an effect of NipA binding to cuticle components or by alternating gene expression of *C. elegans*.

Gene expression analysis of nematodes expressing *nipA* showed upregulation of genes involved in the stress response, detoxification, or pathogen recognition. Together with the observation of the localization of ubiquitously expressed NipA-mCherry in the coelomocytes' vacuoles, this suggests active degradation of NipA that could explain the induction of detoxification gene transcription[42]. Notably, in human injuries, the expression of type I and type III collagens are upregulated to promote wound healing[43]. Hence, it could be that *A. flagrans* inhibits the healing process by downregulating some collagen genes.

Another explanation for the upregulation of the stress response is a secondary effect of damage caused through blister formation. It is known that disruption of some cuticle collagen genes activate distinct stress response pathways. Specifically, disruption of annular furrows activates detoxification, hyperosmotic-, and antimicrobial response genes[44]. Interestingly, electron microscopy of blistered worms revealed that the furrows and alae remain intact during blister formation, so the induced stress responses here might be distinct.

Another interesting observation in the RNAseq analysis of *nipA*-expressing nematodes is the downregulation of some C-type lectins (CLECs). C-type lectins are associated with pathogen recognition, and in vertebrates their expression is upregulated during inflammatory processes. In *C. elegans*, *clec* expression is pathogen-dependent[45]. One explanation for the downregulation in *nipA*-expressing mutants could be an active preparation for the impending infection by *A. flagrans*. It is also described that CLEC genes like CLEC-60 are upregulated by so called DAMPs or damage-associated molecular patterns. In this theory, surveillance pathways will detect pathogens by sensing disruptions in cellular processes or structures that can be caused by virulence factors[46]. In this manner, the blister formation caused by NipA could lead to the release and recognition of DAMPs which induces immune responses. The disruption of cuticle integrity can trigger immune response signaling cascades, similar to what is observed with exotoxin A-induced translation inhibition during *Pseudomonas aeruginosa* infections in *C. elegans*[47]. However, little is known about the regulation of *clec* gene expression in *C. elegans*, making it difficult to make conclusions about a possible mechanism of NipA in regulating immunity genes in *C. elegans*. Another possibility for the observed differential expression in immunity genes is that expression of NipA leads to changes in core cellular processes which are detected by the so-called surveillance immunity of the nematode. This is a concept related to effector-triggered immunity in plants where hosts detect pathogens by surveillance of the homeostasis of key physiological processes, like the proteasome, translation, or mitochondrial function[48]. It is still not clear if the differential gene expression is a primary effect of NipA itself or a secondary effect by the damage caused by NipA.

Analyzing the NipA sequence with AlphaFold revealed similarities to proteins containing EGF domains, which are typically found in the extracellular regions of membrane-bound proteins[49–51]. In *C. elegans*, many EGF domain-containing proteins are localized in the epidermal membrane. For instance, MUP-4 and MUA-3 proteins connect the ECM of the cuticle to muscles using fibrous organelles (FOs)[52]. FOs resemble hemidesmosomes in vertebrates, in which mutations have been linked to blistering skin diseases[53,54]. The potential interaction of NipA with MUA-3 or MUP-4 could disrupt the integrity of the cuticle-epidermis connection, favoring blister formation. However, this effect might be subtler during the actual interaction between the fungus and nematode, possibly limited to a few cells and not leading to visible blisters but facilitating the penetration of fungal hyphae. Yeast two-hybrid studies were performed to investigate interactions between FO components and NipA. Although we detected potential interactions between NipA and MUP-4 as well as VAB-10A, negative controls for VAB-10A and MUP-4 showed autoactivation for both proteins, and hence no conclusion could be drawn.

Another potential target of interacting proteins could be the EGF receptors (EGFRs) themselves. EGFR signaling is crucial in regulating growth, cell division, and differentiation in mammalian cells[55]. The EGFR pathway has also been linked to *P. aeruginosa* infections in *C. elegans*, where the EGFR pathway activates the signal-regulated kinase (ERK) pathway. LIN-3, an EGF-like protein, acting via LET-23, activates the LET-60/MPK-1 signaling cascade. This suggests that the EGFR pathway plays a significant role in both *C. elegans* development and its immune response[56].

Besides those possibilities, cuticle collagen itself could be the target of NipA (Fig. 10). Blister formation is a morphological feature

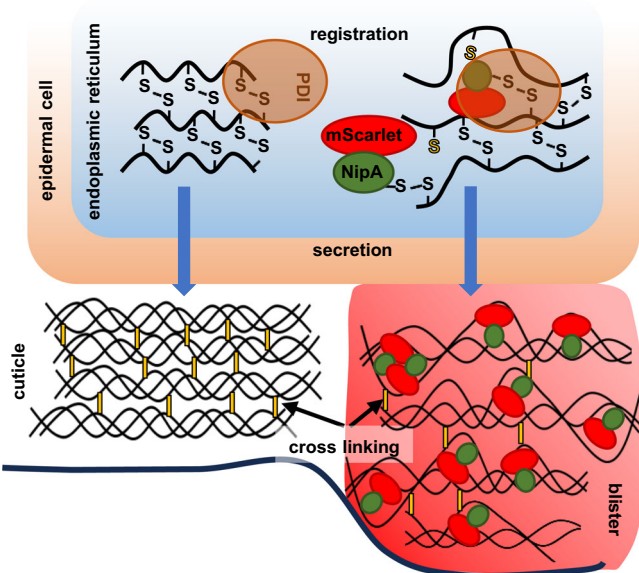

**Fig. 10 | Working model for NipA induced blistering.** During registration of single collagen strains NipA binds to cysteines of collagen by its C-terminal unbound cysteine catalyzed by PDI (protein disulfide-isomerase). Binding of NipA interferes with the trimerization of the collagens leading to a weakening of the cuticle.

observed exclusively in adult nematodes[4]. This observation can be explained by the structural composition of the adult cuticle, which is the only one that possesses a fluid-filled space between the cortical and basal layers[57]. Some blister phenotypes are attributed to mutated collagens, as in BLI-1 and BLI-2 mutants[58,59]. Incorrect processing of collagens due to mutations in enzymes like BLI-3 or BLI-4 can also lead to such phenotypes[3,60]. The fact that *nipA* expressed under the *col-19* promoter in adult animals' epidermis leads to blister formation, and the expression of *nipA* in embryos and early larval stages under the *eft-3* and *dpy-7* promoters results in deformed larvae, suggests an effect of NipA on cuticular components. The blister phenotype was stronger if NipA was secreted and if NipA was fused to *mScarlet*. This suggests that indeed the cuticle components itself are the targets and that NipA may interfere with the integrity of the cuticle. This could be due to the physical presence of the protein. NipA could bind to a target protein and compete with other interactions of that protein. If NipA is artificially enlarged through the fusion with the fluorescent protein, the hinderance effect is apparently more severe. A candidate for the interaction of NipA with a target protein is the first cysteine residue in the protein sequence. The structural predictions suggest that it is not involved in any intramolecular disulfide bridges and thus could interact with other cysteine- or methionine-containing target proteins to form intermolecular bridges. Likewise, mutagenesis of Cys-23 rendered NipA non-functional in terms of blister induction. Re-complementation of the *nipA*-deletion strain with *nipA* with mutated Cys-23 failed to rescue the penetration phenotype, pointing out a potential function of Cys-23 during the penetration process. The structure of collagens is characterized by interrupted blocks of Gly-X-Y sequences flanked by conserved cysteine clusters which may be involved in the registration events by protein disulfide isomerase (PDI) in the collagen biosynthesis pathway[61–63]. The free SH-group of cysteine could form a disulfide bridge with the SH group of cysteines of the collagens and inhibit the correct assembly of collagen trimers. Mutation of the first cysteine in NipA could therefore prevent this binding so that collagens register and trimerize normally leading to no blistering (Fig. 10). It remains to be shown how NipA affects collagen during the infection process. On the other hand, proteins containing EGF domains may also be a target of NipA.

In summary, this study revealed that NipA plays a role during *A. flagrans* penetration of *C. elegans*. Although the formation of NipA-induced blisters is to some extent artificial due to excessive NipA production, it clearly shows potential weakening of the cuticle in the natural infection process.

## Methods

### Strains and culture conditions
Cultivation of organisms: *A. flagrans* (CBS 349.94) was obtained from the CBS-KNAW culture collection (The Netherlands) and was cultured at 28 °C on potato dextrose agar (PDA). *C. elegans* cultivation and synchronization was performed according to the worm book. Strains are listed in Supplementary Information Table 1.

### Trap induction of *A. flagrans*
For trap induction spores *of A. flagrans* were inoculated on low-nutrient-agar[19] (KCL 1 g/L, MgSO$_4$-7H$_2$O 0.2 g, MnSO$_4$-4H$_2$O 0.4 mg, ZnSO$_4$-7H$_2$O 0.88 mg, FeCl$_3$-6H$_2$O 3 mg, Agar 10 g, pH 5.5) and co-incubated with *C. elegans* at 28 °C in darkness for at least 12 h.

### Generation of transgenic microorganisms
Protoplast transformation of *A. flagrans*: Protoplast transformation was carried out as described[19]. $6 \times 10^6$ protoplasts were transformed with 6–7 μg of DNA. Transformants were incubated at 28 °C for 4–7 days on PDA supplemented with 100 μg/ml hygromycin-B or 150 μg/ml geneticin (G418) and/or 100 μg/ml nourseothricin.

Microinjection of *C. elegans*: To generate transgenic *C. elegans* strains expressing extrachromosomal arrays, a DNA mix of 5 ng/μl linearized plasmid harboring the desired transgene, 5 ng/μl pharyngal marker plasmid and 140 ng/μl 1 kb DNA Ladder (Eurofins) as filler, was injected into the gonads of young adult worms. Co-injection marker-positive transformants were selected.

### Plasmid construction
All plasmids are listed in Supplementary Information Table 2 and oligonucleotides in Supplementary Information Table 3. For the laccase assay, the modified vector pOF018 was used for the expression of the *A. nidulans* lccC gene (AspGD identification AN5397) under the constitutive *A. nidulans* glyceraldehyde-3-phosphate dehydrogenase (*gpdA*) promoter as described[19]. The *nipA* gene sequence was fused to the 3′ end of the laccase C by insertion into the vector backbone using the restriction enzymes *Asc*I and *Age*I. The control plasmid was built using Gibson assembly. Standard protocols were used for *Escherichia coli* transformation and plasmid isolation[64].

For the promoter fusion 1.7 kb upstream of the ORF of *dfl_005407* were amplified with overhangs in the pVW23 vector. The backbone was amplified using primers binding outside of the *h2b* promoter. Both amplicons were assembled in a 3:1 ratio using Gibson assembly.

Plasmids for NipA localization were constructed using the Gibson assembly method. The *nipA* promoter was amplified together with the *nipA* ORF with gDNA as template and cloned into a modified version of the pNH21 vector. For the NipA-mCherry fusion pNH53 was used as backbone. The *nipA* promoter and ORF were amplified from gDNA with overhangs and cloned into the backbone. *mCherry* was amplified from pJM10 with overhangs and cloned into the *nipA* ORF and the *gluC* terminator.

For heterologous expression in *C. elegans* the ORF of *nipA* was amplified with or without signal peptide (AA 1-16) from cDNA and cloned into a modified version of the pPF37 vector using Gibson assembly. For the localization of the constructs NipA was tagged C-terminally with *mScarlet*. The constructs were expressed under the all-tissue *eft-3* promoter or the hypodermal *dpy-7* or *col-19* promoters.

### Laccase assay
The Laccase-assay was performed as described[19,20]. Mycelium was transferred to a PDA plate containing 1 mM ABTS using a toothpick. Plates were incubated for 48 h at 28 °C. Laccase activity was correlated with blue ABTS• formation.

### Southern hybridization
Southern blot analysis was performed for verification of the NipA deletion. Isolated DNA of *nipA*-deletion mutants and wild-type was isolated and digested overnight with *Nco*I (New England Biolab, Frankfurt). Analysis was carried corresponding to standard protocol. The labeling probe was maintained by PCR amplification of the right border of NipA using specific primers and DIG-labeled dNTPs (Roche, Mannheim). DNA was blotted on a nylon membrane (Roti-Nylon plus, Roth, Karlsruhe). Membrane was incubated with a 1:10,000 dilution of Anti-Digoxigenin-AP Fab Fragment (REF 11093274910; Roche, Mannheim). For development of the membrane CDP-Star solution (Roche, Mannheim) was added directly before exposure in a Chemi-Smart chemiluminescence -system (Peqlab).

### RNA extraction and quantitative RT-PCR
For RNA extraction $10^6$ *A. flagrans* spores were plated on cellophane covered LNA plates and incubated for 48 h at 28 °C for the uninduced samples. For the induced samples a mixed population of *C. elegans* was added after the first 24 h and co-cultivated for another 24 h at 28 °C. Trap formation was microscopically observed, the mycelium was collected and immediately frozen in liquid nitrogen. A micro pestle was used to grind the material. Total RNA was isolated with Trizol reagent (Invitrogen, Karlsruhe, Germany). DNase digestion was performed using the Turbo DNA-free Kit (Invitrogen, Karlsruhe, Germany). The qRT-PCR was performed using the Luna Universal One-Step RT-qPCR KIT (NEB) on a CFX Connect Real-Time PCR Detection System (Bio-Rad, Munich, Germany). Each reaction mix contained 0.2 μM oligonucleotides and 100 ng of RNA in a total volume of 20 μl. The gamma actin orthologue DFL_002353 was used as an internal reference gene for normalization for *A. flagrans* and the *actin-1* gene (WBGene00000063) was used for normalization in *C. elegans*. The experiment was carried out in technical and biological triplicates. For RNA sequencing RNA was sent to BGI (Honkong, China). The gene IDs obtained from BGI were transformed into WormBase gene IDs using g:convert Gene ID conversation. Genes without WormBase ID were excluded from further analysis.

### Penetration assay
To investigate the dynamics of trap formation and nematode penetration, spores of the respective strains were transferred onto thin LNA slides. A mixed population of N2 nematodes was then added to induce trap formation, followed by incubation at 28 °C for 24 h. Prior to microscopy analysis, untrapped worms were gently washed off with dH$_2$O, and a synchronized population of young adult N2 nematodes was added to the slides. Long-term imaging was performed using a Zeiss AxioObserver.Z1 microscope equipped with a multi-laser module, a spinning disc module (CSU-X1M 5000), and an evolve 512 camera. Several positions with empty traps were saved. The experiment was set to run for 11 h, with images captured every 9 min. Pictures were taken in brightfield and RFP channels. The penetration time was determined by measuring the duration from the moment a nematode entered the trap until the accumulation of CyrA-mCherry was observed.

### Statistics and reproducibility
Group sizes are described in the figure legends. Unless specifically noted, each experiment was repeated three or more times independently. Data were collected from three biological repeats unless

otherwise noted. Data shown in graphs or plots represent mean ± the standard deviation (SD), as indicated in the figure legends. Plotted data points are shown. Details were given in the above methods and source data files. Data diagrams and statistical analyses were made with the software GraphPad Prism 8.0. For statistical analysis, a two-sided unpaired $t$-test was performed. The statistical significance is considered as the $P < 0.05$.

## Protein domain prediction and AlphaFold 2

NCBI protein database was used to obtain the NipA protein sequence (RVD87166.1). SignalP5.0 was used for signal peptide prediction and WolF PSORT for localization prediction in the host. Protein domain was predicted by ProtNLM of UniProt. NipA was modeled using AlphaFold 2.3.1 for structural predictions[30,65]. Modeling and color coding of AlphaFold 2 predictions were done with PyMOL 2.4.2.

## Microscopy

Spores were inoculated onto a thin LNA slide for microscopic analysis and incubated for at least 12 h at 28 °C. Conventional fluorescence images were captured at room temperature using a Zeiss Plan-Apochromat 63x/1.4 Oil DIC, EC Plan-Neufluar 40x/0.75, Es Plan-Neofluar 20x/0.50, or EC Plan-Neofluar 10x/0.30 objective with a Zeiss AxioImager Z.1 and Axio-CamMR. For Confocal microscopy, the Zeiss LSM 900 with Airyscan 2 was used. Images were collected using ZEN 2012 Blue Edition.

Cryo-Scanning electron microcopy (Cryo-SEM) was performed as described by ref.[66]. For fungal samples the fungus was inoculated on thin LNM agar slides and co-incubated with *C. elegans* for 24 h at 28 °C. One hour prior microscopy untrapped nematodes were washed off and synchronized young adult nematodes were added. For microscopy areas with trapped nematodes were selectively cut out and fixed on the transfer shuttle with conductive mounting medium.

Stereomicroscopy was performed using a Zeiss Lumar.V12 with an AxioCam HRc and NeoLumar S 1.5x objective. Images were collected with the AxioVision software.

For long-term observation penetration assays of *A. flagrans* wild-type and the *nipA*-deletion strain, an AxioObserver Z1 inverted microscope with a 10x/0.30 N.A. objective (Zeiss) was used.

The fungal cell wall was visualized by Calcofluor-white (CFW, fluorescent brightener 28, Sigma Aldrich) as described[19].

## Reporting summary

Further information on research design is available in the Nature Portfolio Reporting Summary linked to this article.

# Data availability

All data generated or analyzed during this study are included in this published article or source data file. Source data are provided with this paper. The *Arthrobotrys flagrans* genome database used in this study is available at the National Center for Biotechnology Information Gen-Bank under the accession number PRJNA494930. References to this accession number can be found throughout this paper. RNAseq data are deposited here: https://www.ncbi.nlm.nih.gov/biosample/41737189 https://www.ncbi.nlm.nih.gov/biosample/41737190 https://www.ncbi.nlm.nih.gov/biosample/41737191 https://www.ncbi.nlm.nih.gov/biosample/41737192. Source data are provided with this paper.

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

## Acknowledgements

We thank Valentin Wernet and Natalia Requena for fruitful discussions. The work was supported by the *Deutsche Forschungsgemeinschaft* (DFG Fi459/26-1) and the Karlsruhe Institute of Technology (KIT).

## Author contributions

J.E. performed most of the experiments and drafted the manuscript. N.W. did some initial experiments and supervised J.E. at the beginning of the project. B.H. did the SEM analysis and pictures. E.W. was responsible for injections and generation of transgenic *C. elegans* strains. R.F. supervised and financed the project and edited the manuscript.

## Funding

## Competing interests

The authors declare no competing interests.
