## [Peer Review File · Nature Communications]

The cysteine-rich virulence factor NipA of *Arthrobotrys flagrans* interferes with cuticle integrity of *Caenorhabditis elegans*REVIEWER COMMENTS

Reviewer #1 (Remarks to the Author):

The paper discusses the role of Small-secreted proteins (SSPs) as crucial effectors in the interaction between nematode-trapping fungi (NTF) and nematodes. Specifically, it characterizes a nematode-induced protein (NipA) as a key SSP. The research highlights that NipA exhibits transcriptional upregulation in fungal traps and accumulates at the penetration site. Moreover, it reveals a decelerated penetration rate after deletion of *nipA* when compared to the wild type. Further mechanistic studies demonstrate that the expression of NipA within the epidermis of *C. elegans* leads to aberrant regulation of specific pathways and the formation of characteristic blisters. This study provides valuable insights into the multifaceted role of NipA in the NTF-nematode interaction.

This work builds upon prior research into SSPs of NTF, following the SSPs analysis in *PLoS Genet* 15, 653 e1008029 (2019) and the characterization of the small-secreted cysteine-rich protein *CyrA* in *Arthrobotrys flagrans*, as presented in *PLoS Pathog* 17, e1010028 (2021) by the same research group. The paper is well-structured and effectively presented. The conclusion is robustly supported by the detailed data. However, I have one concern that I would like to discuss with the authors.

The paper highlights that NipA is upregulated in the fungal traps and its encoding protein accumulates at the penetration site, suggesting that NipA is involved in the initial infection process. To enhance clarity, it would be beneficial for the authors to specify whether NipA is primarily confined to the fungal infection peg or if it also enters the epidermis of *C. elegans*. If NipA is primarily localized to the infection peg, the paper should address how NipA influences *C. elegans* by using heterologously expressed NipA and whether the receptor of NipA in *C. elegans* has been determined. At the very least, the authors should discuss this aspect in the text to provide a more comprehensive understanding of NipA's role in the interaction.

Reviewer #2 (Remarks to the Author):

This study presents novel information on the role of *nipA*, a putative effector gene in the nematode parasitic fungus *Arthrobotrys flagrans*. NipA is a small cysteine-rich secreted protein with an EGF-like domain, whose expression is upregulated during infection. The authors conducted fluorescence microscopy analysis to show that NipA protein accumulates at the external region of the infection site. Furthermore, a fungal knockout mutant in the *nipA* gene was slightly delayed in penetration of the nematodes. High expression of *nipA*-mScarlet fusion protein in *C. elegans* under the control of different constitutive or tissue-specific promoters led to a deformed phenotype in some nematode larvae. Furthermore, specific expression of *nipA* in the cuticle led to the formation of fluorescent blister-like structures in some individuals, while expression of a mutated *nipA* allele in which Cys23 was replaced with alanine did not cause blister formation. RNAseq revealed that 238 *C. elegans* genes were differentially expressed upon expression of *nipA*, including downregulation of collagen and C-type lectins, and upregulation of genes associated with stress response, detoxification, or pathogen recognition.

While effectors have been investigated in detail in fungus-plant interactions, there are very few studies on secreted effectors in nematode-pathogenic fungi. To my knowledge, this is one of the few studies where the role of a secreted effector during infection has been characterized in detail. Therefore, this work is of high interest for the research community.

In general, the experiments are thoroughly performed and the results are solid. In particular, the microscopy analyses are of high quality. However, there are some weaknesses in the functional analyses, particularly in the experiments done with the fungal knockout mutant (see details below).

Moreover, as a non-specialist in nematode biology, I have some doubts on the significance of some

of the phenotypes associated with nipA overexpression in *C. elegans*. For instance, the authors observed a deformed phenotype in 15-20% of the larvae overexpressing nipA compared to 5% in control larvae expressing only mScarlet. Is this difference biologically meaningful. Why is this phenotype only observed in such a low fraction of the larvae and not in the rest? Similar questions apply to the blister formation. The authors themselves concede in the discussion that the formation of NipA-induced blisters could be, to some extent, artificial due to excessive NipA production.

In my opinion, these questions need to be discussed and clarified before the article can be accepted. Furthermore, the points listed below need to be addressed.

Major point:

Line 310 and following: While these results suggest that penetration of the nematode is slightly delayed in the nipA mutant, the evidence is not very robust because (1) apparently only a single knockout mutant was tested and (2) no complemented strains of this knockout mutant were generated to exclude additional off-target effects. To corroborate these results, complementation with the wt nipA allele and, ideally, with the cys23ala mutant allele as a negative control, needs to be performed in the fluorescent CyrA-mCherry reporter strain, and nematode infection studies with the complemented strains need to be done.

As a further point, besides measuring penetration time, did the authors test whether delayed penetration in the nipA mutant has any effect on the severity of the disease or the speed of killing? Such quantitative results would strongly support a role of nipA in infection.

Minor points:

Line 74: "the fungus produces arthrospores to inhibit trap formation" This sentence sounds counterintuitive, since in this paragraph the authors describe the fungal processes that promote predation of nematodes. Maybe attach this sentence after, e.g. ... on the other hand, under non-inducing conditions of the fungus produces arthrospores to inhibit trap formation...

Line 242: The results section directly starts with a chapter on the characteristics of the NipA protein, without mentioning why exactly this protein was selected for analysis among the several hundred predicted effector candidates. Were other candidate effectors tested before selecting this one? I suggest to add a short paragraph explaining this point.

Fig. 2A: Explain "relative expression" (versus actin?) in Fig legend or directly on the y axis.

Fig. 2B,C: I suggest to indicate "Ctrl" and "induced" above each graph to make the figure more self-explanatory.

Line 340 and following: please specify in the text for each of the transformed nematode strains (KIT numbers) whether they were generated in the N2 or the GS2478 background.

Reviewer #3 (Remarks to the Author):

Review of NCOMMS-23-46907 – Muntasir Kamal

The authors made an attempt to characterize an SSP (small secreted protein) from the nematode trapping fungus *A. flagrans* that they call NipA (nematode induced protein). The authors show by using a fluorescent tagged protein that accumulates internally later in the infection process that the infection process is slowed when NipA is deleted. It is further shown that heterologous expression of NipA in *C. elegans* causes the formation of blisters in the adult cuticle, with other results indicating NipA accumulates in these blisters and may drive blister formation. The authors furthermore perform global RNA expression analyses in NipA expressing and control animals and found immunity and stress response genes are upregulated while genes involved in stress

response and ECM production (e.g., collagens) are downregulated.

Overall, this is an important work because it adds to our understanding of host-pathogen interactions in the model *C. elegans* which is still a relatively small and young field. The experiments are well-planned and properly controlled. I would, however, have liked to see some more experiments done. In particular- SEM images of worms in early stages of infection by *A. flagrans* and in the presence of the NipA-deleted strain. This would also show what types of deformations occur (such as blisters or blister-like objects) on the outer layers of the cuticle in the presence and absence of NipA. A big part of the paper is about the blister phenotype produced when NipA is expressed. This is despite the fact that it was not clear whether blisters are part of the actual infection process or artefactual due to heterologous expression of a (fluorescent-tagged) foreign protein. Even if blisters are caused by NipA affecting differential gene expression (or by other means) as the authors propose, given the observation that the site of NipA action is the outer cuticle and occurs early during infection calls into question whether the blisters are relevant in the natural infection process. Perhaps an alternative experiment would have been to express and purify the NipA protein and expose the worms to purified NipA and testing whether that produces symptoms of infection. Pathogen extracts (e.g., Grover et al., 2021 (PMID: 34722333)) derived from animal-fungal populations with NipA(wt) and NipA deleted strains could be administered to worms as an alternative to purified protein.

The writing deserves some reorganization and re-phrasing throughout. Regarding organization, results described in lines 378-380 are more interesting/exciting than reduced viability that was seen in both strains, so I would mention the more interesting result first. Also, in line 398, after presenting results with the NipA expressing strain, the reader naturally wonders, did the mScarlet only expressing strain give any blisters? Which was not addressed until much later, in lines 413-414.

In addition, several places have awkward phrasing that should be fixed. Every figure needs to be properly referenced in-text. Wherever quantified data are being presented and being compared between samples, proper stats (mean +/- SD, p-value) all need to be mentioned. Strain names in brackets are not needed in-text, and something like "see methods" should be sufficient. The corresponding figure must contain quantified data with error bars. Pictures are nice, but the quantified data must be included along with the pictures.

I thought the entire data with GS2478 strain could be removed from the paper. I could not see the relevance of the strain other than the stronger coelomocyte accumulation of NipA, but since the consequence of this accumulation is not sufficiently understood/addressed, the whole utility of GS2478 is not clear in the paper. For example, following the coelomocyte result the authors seem to revert to the xN2 strain for viability assay (lines 363-366) (Fig 5E).

Regarding the morphological defect in the NipA-expressing larvae, I am not fully convinced that the morphological deformations seen in early larvae are due to NipA targeting the cuticle. The larval effect is not sufficiently described. For example, it is not clear whether the deformation is due to a direct effect of NipA on the cuticle/hypodermis or the foreign protein has non-specific toxic effects.

Also regarding the morphological defect- did the authors see similar morphological defects in any young adults/adults that escaped the early larval sickness/lethality? If yes, why did the authors not see any effect of NipA expression, especially given they did the nematode trapping + infection assay with synchronized young adults?

I feel the morphology study deserves a control where some other protein is expressed hypodermally together with mScarlet under the same promoter, e.g. *dpy-7p::DPY-7::mScarlet* vs. *dpy-7p::NipA::mScarlet*.

Line 247-248: Please fix

Line 257: Please fix

Line 261: Please fix

Line 357: What "potent effect" did the authors expect to get? Did the authors end up seeing more potent effect?

Line 367: "caused such a strong phenotype" – subjective, must be modified.

Line 375: What was the reason for seeing dumpy worms in the control (mScarlet only) strain?

Line 397: "some nematodes" – what percentage of worms?

Line 407: "This suggests that the NipA might leak into the ECM" – or maybe it is cleaved mScarlett only – but this seems to be addressed later, in line 438 (Fig 7A).

Line 412: "blistering was delayed"- but this would suggest an effect of the fluorophore?

Line 427: "even after several days" – please mention precisely after how many days (mean +/-

SD).

Line 476: Figure 8C network cluster map needs to be in better resolution.

Line 485: "our results revealed that NipA is responsible for weakening.." – I would tone it down. I don't think that the DEG data suggest function of NipA- for that functional assays must be carried out.

Lines 502-506: Discussion on blister phenotype. All this discussion would be relevant and more interesting if the blisters (or something similar) were shown to be part of natural *A. flagrans* infection of *C. elegans*.

Line 513-514: "Hence, it could be that *A. flagrans* inhibits the healing process by downregulating some collagen genes." – but it could be the worm's self-defense response to downregulate cuticle components that harbor the target(s) of NipA? For example, see Osman et al., 2018 (PMID: 29398216).

Lines 552-554: "...this effect might be subtler during the actual interaction between the fungus and nematode, possibly limited to a few cells and not leading to visible blisters but facilitating the penetration of fungal hyphae." – Would be interesting to show SEM shots of early infected worms.

Missing in-text Figure references:

Lines 260, 374, 376, 380, 428

Missing statistics:

Lines 292, 333, 334

Awkward phrases:

Line 275: because overall fungal material from induced plates was pooled for RNA extraction.

Line 278: The fusion protein will accumulate

Line 319: because the lack of valid parameters for penetration

Line 403: NipA locates

Line 405: blister formation was still possible

Line 417: according to the literature

Line 423: (typo) und

Line 424: *C. elegans* needs italics

Line 459: using the TRIZOL method and sent for RNA sequencing.

Line 462: Line 462: WormCat was used for a gene ontology analysis of the shared up- and downregulated gene sets³³ – awkward placement of sentence

Line 559: Another potential target of interacting proteins

Figure 3C: Check for autofluorescence, which also appears green. Show control panels for autofluorescence.

Reviewer #1:

The paper discusses the role of Small-secreted proteins (SSPs) as crucial effectors in the interaction between nematode-trapping fungi (NTF) and nematodes. Specifically, it characterizes a nematode-induced protein (NipA) as a key SSP. The research highlights that NipA exhibits transcriptional upregulation in fungal traps and accumulates at the penetration site. Moreover, it reveals a decelerated penetration rate after deletion of *nipA* when compared to the wild type. Further mechanistic studies demonstrate that the expression of NipA within the epidermis of *C. elegans* leads to aberrant regulation of specific pathways and the formation of characteristic blisters. This study provides valuable insights into the multifaceted role of NipA in the NTF-nematode interaction.

This work builds upon prior research into SSPs of NTF, following the SSPs analysis in PLoS Genet 15, 653 e1008029 (2019) and the characterization of the small-secreted cysteine-rich protein CyrA in *Arthrobotrys flagrans*, as presented in PLoS Pathog 17, e1010028 (2021) by the same research group. The paper is well-structured and effectively presented. The conclusion is robustly supported by the detailed data. However, I have one concern that I would like to discuss with the authors.

The paper highlights that NipA is upregulated in the fungal traps and its encoding protein accumulates at the penetration site, suggesting that NipA is involved in the initial infection process. To enhance clarity, it would be beneficial for the authors to specify whether NipA is primarily confined to the fungal infection peg or if it also enters the epidermis of *C. elegans*.

NipA is predicted to be secreted and to localize in the extracellular space of the host (WoLFP SORT). Secretion was confirmed experimentally. However, we were unable to detect NipA inside the nematode. This is due to the dilution of the protein. Even other virulence proteins studied in our lab which enter the *C. elegans* cells and nuclei cannot be followed by conventional fluorescence microscopy. From our experience with other filamentous fungi like *A. nidulans* it is very difficult to detect secreted proteins outside the fungal cell.

If NipA is primarily localized to the infection peg, the paper should address how NipA influences *C. elegans* by using heterologously expressed NipA and whether the receptor of NipA in *C. elegans* has been determined. At the very least, the authors should discuss this aspect in the text to provide a more comprehensive understanding of NipA's role in the interaction.

Heterologous expression of NipA enhances the effect of NipA compared to the natural infection process, given that NipA is expressed in all epidermal cells as compared to localized action around the penetration site. We propose very local action of NipA during the infection process without extensive blister formation.

We tested several possible NipA interaction partners using Y2H approaches. However, no clear results were obtained. If our hypothesis is right that NipA acts on the disulfide bridges of the collagen fibres, a protein target is not expected.

We discussed this aspect.

Reviewer #2:

This study presents novel information on the role of nipA, a putative effector gene in the nematode parasitic fungus *Arthrobotrys flagrans*. NipA is a small cysteine-rich secreted protein with an EGF-like domain, whose expression is upregulated during infection. The authors conducted fluorescence microscopy analysis to show that NipA protein accumulates at the external region of the infection site. Furthermore, a fungal knockout mutant in the nipA gene was slightly delayed in penetration of the nematodes. High expression of nipA-mScarlet fusion protein in *C. elegans* under the control of different constitutive or tissue-specific promoters led to a deformed phenotype in some nematode larvae. Furthermore, specific expression of nipA in the cuticle led to the formation of fluorescent blister-like structures in some individuals, while expression of a mutated nipA allele in which Cys23 was replaced with alanine did not cause blister formation. RNAseq revealed that 238 *C. elegans* genes were differentially expressed upon expression of nipA, including downregulation of collagen and C-type lectins, and upregulation of genes associated with stress response, detoxification, or pathogen recognition.

While effectors have been investigated in detail in fungus-plant interactions, there are very few studies on secreted effectors in nematode-pathogenic fungi. To my knowledge, this is one of the few studies where the role of a secreted effector during infection has been characterized in detail. Therefore, this work is of high interest for the research community.

In general, the experiments are thoroughly performed and the results are solid. In particular, the microscopy analyses are of high quality. However, there are some weaknesses in the functional analyses, particularly in the experiments done with the fungal knockout mutant (see details below).

Moreover, as a non-specialist in nematode biology, I have some doubts on the significance of some of the phenotypes associated with nipA overexpression in *C. elegans*. For instance, the authors observed a deformed phenotype in 15-20% of the larvae overexpressing nipA compared to 5% in control larvae expressing only mScarlet. Is this difference biologically meaningful.

We added the quantification in Fig. 5 F.

Why is this phenotype only observed in such a low fraction of the larvae and not in the rest? Similar questions apply to the blister formation. The authors themselves concede in the discussion that the formation of NipA-induced blisters could be, to some extent, artificial due to excessive NipA production.

The variability can be explained by different expression levels in different animals because the expression of the constructs in DNA arrays. If higher NipA concentrations are lethal, only animals with low or medium expression level will be observable.

In my opinion, these questions need to be discussed and clarified before the article can be accepted. Furthermore, the points listed below need to be addressed.

We were very thankful for these points because this paper will be relevant for two scientific communities, the fungal and the *C. elegans* community. Therefore, it is very important to clarify the methods for both. For fungal people it sounds rather odd that individuals have different expression levels due to the expression in DNA arrays. We discussed the concerns in the text and hope that now also people not working with *C. elegans* can follow our arguments and results.

Major point:

Line 310 and following: While these results suggest that penetration of the nematode is slightly delayed in the *nipA* mutant, the evidence is not very robust because (1) apparently only a single knockout mutant was tested and (2) no complemented strains of this knockout mutant were generated to exclude additional off-target effects. To corroborate these results, complementation with the wt *nipA* allele and, ideally, with the *cys23ala* mutant allele as a negative control, needs to be performed in the fluorescent CyrA-mCherry reporter strain, and nematode infection studies with the complemented strains need to be done.

We added the re-complemented strains. See Fig. 4D.

As a further point, besides measuring penetration time, did the authors test whether delayed penetration in the *nipA* mutant has any effect on the severity of the disease or the speed of killing? Such quantitative results would strongly support a role of *nipA* in infection.

If penetration in the *nipA*-deletion strain is delayed, the whole process will be slower than during the infection with wild type. The paralysis time itself for instance was not affected.

Minor points:

Line 74: “the fungus produces arthrospores to inhibit trap formation” This sentence sounds counterintuitive, since in this paragraph the authors describe the fungal processes that promote predation of nematodes. Maybe attach this sentence after, e.g. ... on the other hand, under non-inducing conditions of the fungus produces arthrospores to inhibit trap formation...

Done

Line 242: The results section directly starts with a chapter on the characteristics of the NipA protein, without mentioning why exactly this protein was selected for analysis among the several hundred predicted effector candidates. Were other candidate effectors tested before selecting this one? I suggest to add a short paragraph explaining this point.

We added a paragraph explaining how we chose to analyse NipA.

Fig. 2A: Explain “relative expression” (versus actin?) in Fig legend or directly on the y axis.

Done

Fig. 2B,C: I suggest to indicate “Ctrl” and “induced” above each graph to make the figure more self-explanatory.

Done

Line 340 and following: please specify in the text for each of the transformed nematode strains (KIT numbers) whether they were generated in the N2 or the GS2478 background.

Information of the strain backgrounds are mentioned either before or after the paragraph regarding the generated lines. Following reviewer 3, the strain names in brackets were deleted.

Reviewer #3,4:

Review of NCOMMS-23-46907 – Muntasir Kamal

The authors made an attempt to characterize an SSP (small secreted protein) from the nematode trapping fungus *A. flagrans* that they call NipA (nematode induced protein). The authors show by using a fluorescent tagged protein that accumulates internally later in the infection process that the infection process is slowed when NipA is deleted. It is further shown that heterologous expression of NipA in *C. elegans* causes the formation of blisters in the adult cuticle, with other results indicating NipA accumulates in these blisters and may drive blister formation. The authors furthermore perform global RNA expression analyses in NipA expressing and control animals and found immunity and stress response genes are upregulated while genes involved in stress response and ECM production (e.g., collagens) are downregulated.

Overall, this is an important work because it adds to our understanding of host-pathogen interactions in the model *C. elegans* which is still a relatively small and young field. The experiments are well-planned and properly controlled. I would, however, have liked to see some more experiments done. In particular- SEM images of worms in early stages of infection by *A. flagrans* and in the presence of the NipA-deleted strain. This would also

show what types of deformations occur (such as blisters or blister-like objects) on the outer layers of the cuticle in the presence and absence of NipA.

We added more SEM images. See new Fig. 7.

A big part of the paper is about the blister phenotype produced when NipA is expressed. This is despite the fact that it was not clear whether blisters are part of the actual infection process or artefactual due to heterologous expression of a (fluorescent-tagged) foreign protein. Even if blisters are caused by NipA affecting differential gene expression (or by other means) as the authors propose, given the observation that the site of NipA action is the outer cuticle and occurs early during infection calls into question whether the blisters are relevant in the natural infection process. Perhaps an alternative experiment would have been to express and purify the NipA protein and expose the worms to purified NipA and testing whether that produces symptoms of infection.

We clarified in the text, that large blister formation is not part of the natural infection process.

We purified NipA expressed in *E. coli* and applied it to *C. elegans*. We did not see any blisters. However, we do not know whether the purified protein is still functional. As we propose, cysteines are important for stabilization of the conformation and one cysteine should remain free to act on the target. It is actually rather unlikely that the protein isolated from *E. coli* fulfils all criteria. Further it is not clear, if NipA can pass the glycoprotein-rich surface coat and epicuticle to get to its place of action. In the natural infection process even earlier-secreted effectors (or lytic enzymes) might permeabilize these layers to allow NipA to enter. In turn, NipA could weaken the cuticle more to support the action of other effectors or enzymes.

Heterologous expression of NipA in *E. coli*. Elution buffer was changed in E2 and E3 from Buffer A to M9 buffer to reduce toxicity for Nematodes.

Pathogen extracts (e.g., Grover et al., 2021 (PMID: 34722333)) derived from animal-fungal populations with NipA(wt) and NipA deleted strains could be administered to worms as an alternative to purified protein.

The reviewer suggests an interesting method to purify NipA for application to *C. elegans*. The problem is, that NipA is only expressed in the traps of *A. flagrans* or in the penetration peg and therefore not very abundant.

The writing deserves some reorganization and re-phrasing throughout. Regarding organization, results described in lines 378-380 are more interesting/exciting than reduced viability that was seen in both strains, so I would mention the more interesting result first.

We did some rephrasing and re-organization.

Also, in line 398, after presenting results with the NipA expressing strain, the reader naturally wonders, did the mScarlet only expressing strain give any blisters? Which was not addressed until much later, in lines 413-414.

Added information of mScarlet strains earlier

In addition, several places have awkward phrasing that should be fixed. Every figure needs to be properly referenced in-text. Wherever quantified data are being presented and being compared between samples, proper stats (mean +/- SD, p-value) all need to be mentioned

p-values were added at the points mentioned.

Strain names in brackets are not needed in-text, and something like “see methods” should be sufficient.

Strain names in brackets were removed.

The corresponding figure must contain quantified data with error bars. Pictures are nice, but the quantified data must be included along with the pictures.

The reviewer does not refer to a specific figure. Fig. 2 A-C and Fig. 4 D show images of data that were quantified including the respective data with error bars. We changed graph Fig. 6 D from total blister appearance to blister appearance in the triplicates and added the error bars for standard deviation.

I thought the entire data with GS2478 strain could be removed from the paper. I could not see the relevance of the strain other than the stronger coelomocyte accumulation of NipA, but since the consequence of this accumulation is not sufficiently understood/addressed, the whole utility of GS2478 is not clear in the paper. For example, following the

coelomocyte result the authors seem to revert to the xN2 strain for viability assay (lines 363-366) (Fig 5E).

We think the data of the GS2478 strains expressing strains is still worth to be shown. Since the deformation phenotype and reduced hatching rate were first observed in that strain.

Regarding the morphological defect in the NipA-expressing larvae, I am not fully convinced that the morphological deformations seen in early larvae are due to NipA targeting the cuticle. The larval effect is not sufficiently described. For example, it is not clear whether the deformation is due to a direct effect of NipA on the cuticle/hypodermis or the foreign protein has non-specific toxic effects.

The reviewer raises a good point here. A non-specific toxic effect of the NipA protein can't be fully excluded but the evidence for an effect of nipA during the development of *C. elegans* larvae is based on literature where it is reported that collagen mutations have developmental effects on *C. elegans*. Further, NipA is not the only effector which we are expressing in all stages using the eft-3 promoter, but nipA is the first one showing an effect on young *C. elegans* larvae and embryos.

Also regarding the morphological defect- did the authors see similar morphological defects in any young adults/adults that escaped the early larval sickness/lethality? If yes, why did the authors not see any effect of NipA expression, especially given they did the nematode trapping + infection assay with synchronized young adults?

The virulence assays were not performed with NipA-expressing strains.

I feel the morphology study deserves a control where some other protein is expressed hypodermally together with mScarlet under the same promoter, e.g. dpy-7p::DPY-7::mScarlet vs. dpy-7p::NipA::mScarlet.

We analyzed a strain expressing CyrA as a control.

Line 247-248: Please fix

Done

Line 257: Please fix

Done

Line 261: Please fix

Done

Line 357: What “potent effect” did the authors expect to get? Did the authors end up seeing more potent effect?

We changed the phrasing for clarity.

Line 367: “caused such a strong phenotype” – subjective, must be modified.

Done

Line 375: What was the reason for seeing dumpy worms in the control (mScarlet only) strain?

mScarlet shows some cytotoxicity (cited in the text). Therefore, we compared NipA-mScarlet expressing nematodes with only mScarlet expressing nematodes.

Line 397: “some nematodes” – what percentage of worms?

We changed Fig. 6 D. We now display the average number of blisters in a population of 90 nematodes. The main point is that we never observed blisters in wild type. The number in transgenic nematodes varies because of the expression in DNA arrays.

Line 407: “This suggests that the NipA might leak into the ECM” – or maybe it is cleaved mScarlett only – but this seems to be addressed later, in line 438 (Fig 7A).

Suggestion add in the text

Line 412: “blistering was delayed”- but this would suggest an effect of the fluorophore?

Yes. Therefore, we speculate that NipA acts with the SH group on the integrity of the collagen but that the fluorophore additionally may loosen the mesh.

Line 427: “even after several days” – please mention precisely after how many days (mean +/- SD).

The strain expressing the mutated NipA was observed for 20 days (average lifespan of adult nematodes) but blistering was never observed here.

Line 476: Figure 8C network cluster map needs to be in better resolution.

Done

Line 485: “our results revealed that NipA is responsible for weakening..” – I would tone it

down. I don't think that the DEG data suggest function of NipA- for that functional assays must be carried out.

Done

Lines 502-506: Discussion on blister phenotype. All this discussion would be relevant and more interesting if the blisters (or something similar) were shown to be part of natural *A. flagrans* infection of *C. elegans*.

Since the site of action of effectors is restricted to a very small area in the natural infection, and especially at the penetration site, where pressure is applied to the host, the observation of blisters is not visible (SEM images added). But that does not mean, that NipA has no effect on the nematode cuticle. This effect is enhanced by overexpression of NipA directly in the cuticle of *C. elegans*, giving new insights into the potential function of NipA. This hypothesis is supported by the observation of delayed penetration of the nematode in the fungal *nipA* deletion strain.

Line 513-514: "Hence, it could be that *A. flagrans* inhibits the healing process by downregulating some collagen genes." – but it could be the worm's self-defense response to downregulate cuticle components that harbor the target(s) of NipA? For example, see Osman et al., 2018 (PMID: 29398216).

The reviewer points out an interesting idea. In the literature the reviewer suggested, it is mentioned, that the induction of a *chil* gene could influence the resistance to pathogens in *C. elegans* but nothing is said about the downregulation of specific genes to prevent the infection. We have no evidence which is showing if the downregulation of collagen expression is a primary effect of *nipA* by active expression regulation of the host or if it is a secondary effect from the host to protect itself. But since it is shown in human cells that wounding would induce collagen production, we suggest an active inhibition of collagen production to facilitate the penetration.

Lines 552-554: "...this effect might be subtler during the actual interaction between the fungus and nematode, possibly limited to a few cells and not leading to visible blisters but facilitating the penetration of fungal hyphae." – Would be interesting to show SEM shots of early infected worms.

SEM pictures of fungal WT infecting N2 nematode were added.

Missing in-text Figure references:

Lines 260 citation added, 374 no figure showing that, 376, 380, no pictures to refer to, 428, added.

Missing statistics:

Lines 292

Statistics added

333, 334

SD-values are noted in brackets, p-value for the significant difference between ko and wt samples is added

Awkward phrases:

Line 275: because overall fungal material from induced plates was pooled for RNA extraction.

Changed

Line 278: The fusion protein will accumulate

Done

Line 319: because the lack of valid parameters for penetration

Done

Line 403: NipA locates

Done

Line 405: blister formation was still possible

Done

Line 417: according to the literature

Removed

Line 423: (typo) und

Done

Line 424: *C elegans* needs italics

Done

Line 459: using the TRIZOL method and sent for RNA sequencing.

Phrase removed

Line 462: Line 462: WormCat was used for a gene ontology analysis of the shared up- and downregulated gene sets³³ – awkward placement of sentence

Phrase changed

Line 559: Another potential target of interacting proteins

Changed

Figure 3C: Check for autofluorescence, which also appears green. Show control panels for autofluorescence.

Since the lower panels show much less GFP autofluorescence and a clear CyrA-GFP signal, we deleted the upper panels. The content of the upper and lower panels was somewhat redundant.

REVIEWERS' COMMENTS

Reviewer #1 (Remarks to the Author):

Authors have addressed my concerns and carefully revised the manuscript. However, I feel a little bit confusing about the description of the nematode states, such as "N2", "young adult N2 nematode" etc. through the whole manuscript. I suggest that authors follow the widely accepted description for *C.elegans* life cycle [as described in Clark, L.C., Hodgkin, J. *Caenorhabditis* microbiota: worm guts get populated. *BMC Biol* 14, 37 (2016). <https://doi.org/10.1186/s12915-016-0260-7>] to revise the manuscript.

Reviewer #2 (Remarks to the Author):

In the revised version the authors have satisfactorily addressed all the points raised by the reviewers.

Reviewer #3 (Remarks to the Author):

I have read the author responses and reviewed the revised manuscript. The authors have addressed all my concerns. I have no further questions or concerns.

There was only one comment of referee #1:

Reviewer #1 (Remarks to the Author):

Authors have addressed my concerns and carefully revised the manuscript. However, I feel a little bit confusing about the description of the nematode states, such as "N2", "young adult N2 nematode" etc. through the whole manuscript. I suggest that authors follow the widely accepted description for *C. elegans* life cycle [as described in Clark, L.C., Hodgkin, J. *Caenorhabditis* microbiota: worm guts get populated. *BMC Biol* 14, 37 (2016). <https://doi.org/10.1186/s12915-016-0260-7>] to revise the manuscript.

Response: We would like to keep the nomenclature. It is the same as in the *Wormbook*. It is important not only to write *adult* as in the cited reference, but to highlight that those were *young adult* animals. Otherwise, the nomenclature with L1-L4 larval stages and adult is the same.